# Characterization of a Non-Thermal Plasma Source for the Use as a Mass Spec Calibration Tool and Non-Radioactive Aerosol Charger

Christian Tauber[1], David Schmoll[1], Johannes Gruenwald[2], Sophia Brilke[1], Peter Josef Wlasits[1], Paul Martin Winkler[1], and Daniela Wimmer[1]

[1]Faculty of Physics, University of Vienna, Boltzmanngasse 5, 1090 Vienna, Austria
[2]Gruenwald Laboratories GmbH, Taxberg 50, 5660 Taxenbach, Austria

**Correspondence:** Christian Tauber (christian.tauber@univie.ac.at)

**Abstract.** In this study the charging efficiency of a radioactive and a non-radioactive plasma bipolar diffusion charger (Gilbert Mark I plasma charger) for sub-12 nm particles have been investigated at various aerosol flow rates. The results were compared to classic theoretical approaches. In addition, the chemical composition and electrical mobilities of the charger ions have been examined using an atmospheric pressure interface - time-of-flight mass spectrometer (APi-TOF MS). A comparison of the different neutralization methods revealed an increased charging efficiency for negatively charged particles using the non-radioactive plasma charger with nitrogen as working gas compared to a radioactive americium bipolar diffusion charger. The mobility and mass spectrometric measurements show that the generated bipolar diffusion charger ions are of the same mobilities and composition independent of the examined bipolar diffusion charger. It was the first time that the Gilbert Mark I plasma charger was characterized in comparison to a commercial TSI X-Ray (TSI Inc, Model 3088) and a radioactive americium bipolar diffusion charger. We observed that the plasma charger with nitrogen as working gas can enhance the charging probability for sub-10 nm particles compared to a radioactive americium bipolar diffusion charger. As a result, the widely-used classical charging theory disagrees for the plasma charger and for the radioactive chargers with increased aerosol flow rates. Consequently, in-depth measurements of the charging distribution are necessary for accurate measurements with differential or scanning particle sizers for laboratory and field applications.

## 1  Introduction

Particle properties are influenced by the presence of electrostatic charges, e. g. the deposition of particles in human airways, their collection in a filter and the characterization of size distributions based on the mobility equivalent diameter (Johnson et al., 2020). Consequently, any uncertainty in the charge distribution of aerosol particles propagates into the total measurement uncertainty of electrostatic classifiers (Leppä et al., 2017). The probability that a particle carries one or more charges varies widely over the 1 nm to 1 micrometer size domain, hence, it is crucial to know this probability in order to infer the particle size distribution from the numbers of charged particles that are transmitted through the mobility classifier (Leppä et al., 2017). Bipolar diffusion charging and neutralization is typically done by ionizing radiation, which exposes aerosol particles to high concentrations of positive and negative ions in the carrier gas (Jiang et al., 2014). Subsequent diffusion of the ions brings the

aerosol to a stationary state charge distribution independent of their initial charge state (Cooper and Reist, 1973; Liu and Pui,
1974; Adachi et al., 1985; Reischl et al., 1996). If a high ion concentration and residence time is reached, a charge equilibrium inside the charger leads to a well-known size-dependent charging probability (Fuchs et al., 1965). This stationary state charge distribution is of importance for the use of differential or scanning mobility particle spectrometers, which rely on accurate knowledge of the size-dependent charge fractions (Wang and Flagan, 1990; Jiang et al., 2014).

Aerosol particles below 10 nm in diameter are typically difficult to charge and carry only one electrical charge at maximum
(Wiedensohler, 1988). Quantitative particle detection in this size range is extremely challenging due to high diffusional losses, which results in low number concentrations. Therefore, a higher charging efficiency is of importance to increase the detectable number concentration in the sub-10 nm regime.

The charging of small particles has also become a field of major interest in plasma physics. A vast number of studies about accumulating an electrical charge on dust particles have been published over the last years. Usually grain sizes ranging from
some nm to several $\mu m$ are considered in the experiments and theoretical models. Most of these works were focused on generating plasmas for industrial or space applications (Michau et al., 2016; Deka et al., 2017; Kopnin et al., 2018; Yaroshenko et al., 2018; Intra and Yawootti, 2019). However, one of the most recent developments is the application of plasma in aerosol related topics, such as plasma treatment of aerosol particles (Uner and Thimsen, 2017) and, as a novel topic, charging of aerosol particles or ionization of trace gas compounds (Spencer et al., 2015; Yang et al., 2016; Intra and Yawootti, 2019).
Usually, corona-type discharges are used for the charging purposes in aerosol physics due to their capability of creating high charge densities even at atmospheric pressure. Furthermore, their reproducibility is very high and they are easy to construct and maintain. Low-temperature plasma ionization is known to cause little fragmentation and exhibits a low temperature increase to the surrounding (Harper et al., 2008). The non-thermal plasma in this work is produced by a high frequency generator which is separated by a dielectric barrier to the ground potential (Gruenwald et al., 2015). The term non-thermal plasma is usually
used to describe a discharge in which the electrons are in thermal non-equilibrium with the ions. This means that the average temperature of the gas in such a discharge is far lower than the temperature of a thermal plasma (i.e. some hundred K compared to several thousand K in the latter case). The discharge characteristics can be varied by changing the working gas of the Gilbert Mark I plasma charger (Gruenwald Laboratories GmbH). Plasma discharges in general are on-off devices that combine the simple handling of an X-ray charger with the achievable high ion density of a radioactive americium charger and even higher.
Thus, an atmospheric plasma source is a well-suited device for the ionization process prior to particle number size distribution measurements and mass spectrometric measurements.

In the past, various studies have characterized the charging probabilities and mobility spectra of the ions generated by AC-corona, X-Ray or alpha-radiation based chargers (Wiedensohler et al., 1986; Steiner and Reischl, 2012; Kallinger et al., 2012; Kallinger and Szymanski, 2015). These works revealed deviations in the charge distribution to classical theory, which result in
an increased total measurement error (Johnson et al., 2020).

In this work, we investigated the charging distribution for three different aerosol neutralizer types to reduce the uncertainty under various flow conditions and working gases in the case of the Gilbert Mark I plasma charger. In addition, the chemical

**Table 1.** Flow rates for the aerosol and sheath flow for the used nDMAs with the calculated sheath flow ratio.

| nDMA | Aerosol flow [L/Min] | Sheath flow [L/Min] | Ratio |
|------|---------------------|---------------------|-------|
| 1 | 3.0 | 19.5 | 6.5 |
| 2 | 2.5 | 19.5 | 7.8 |
| 2 | 5.0 | 33.0 | 6.6 |
| 2 | 8.0 | 41.0 | 5.1 |

composition of charger ions of both polarities has been investigated and compared. Furthermore, the optical emission spectra and ozone concentration were measured for the Gilbert Mark I plasma charger.

## 2   Experimental Setup

Here we report on size-dependent charging probability measurements of a non-thermal plasma source (Gilbert Mark I plasma charger, Gruenwald Laboratories GmbH, Austria), an americium 241 ($^{241}$Am) aerosol neutralizer and of a TSI Advanced Aerosol Neutralizer 3088 by means of a tandem DMA (Differential Mobility Analyzer) setup as depicted in Figure 1. Thereby, the charging efficiency of a standard TSI X-Ray neutralizer and an $^{241}$Am aerosol neutralizer could be compared to the plasma source. The atmospheric pressure plasma charger consists of a gas flow that is shielded by another gas flow from the surrounding atmosphere. The plasma is ignited inside the inner flow while the aerosol is administered through the outer gas stream. The main source of the plasma is a high-frequency copper electrode that is situated on the central axis of those two gas streams. According to Kallinger et al. (2012), the used radioactive $^{241}$Am charger has a cylindrical geometry with an axial flow direction. The radioactive source is mounted on the inner wall. The chamber has an inner diameter of about 30 mm and a length of 120 mm. Furthermore, the soft x-ray charger is composed of a stainless-steel tube and a photo ionizer. The aerosol particles are directed along the tube towards the soft x-ray source and leave the charger via an outlet, that is oriented perpendicularly to the axis of the tube. The tube has an inner diameter of 30 mm and a length of 200 mm.

The charging efficiency measurements were performed with sodium chloride and silver nanoparticles at various particle sizes and aerosol flowrates. The nanoparticles were generated with a tube furnace (Carbolite Gero GmbH & Co. KG, Germany) while synthetic air as well as dried and filtered compressed air was used as the carrier gas. An additional dilution flow allowed for controlling the particle concentration of the generated aerosol flow. Downstream of the aerosol generator the nanoparticles were charged with a TSI Advanced Aerosol Neutralizer 3088 and led to a nDMA which was operated as a classifier. A second TSI Advanced Aerosol Neutralizer 3088 neutralized the monodisperse aerosol particles after the nDMA. The geometric standard deviation of the particle size for the used nDMAs was evaluated by Winkler et al. (2008) for a sheath flow of 25 L/Min and an aerosol flow of 4.6 L/Min and is below 1.05 for particles with a mobility diameter down to 2 nm. The resulting flow ratio (sheath / aerosol = 5.4) is close to our measurement with 8 L/Min aerosol flow and the signal-to-noise ratio was comparable to the measurements with lower aerosol flow rates. The different flow settings for the nDMAs are listed in Table 1.

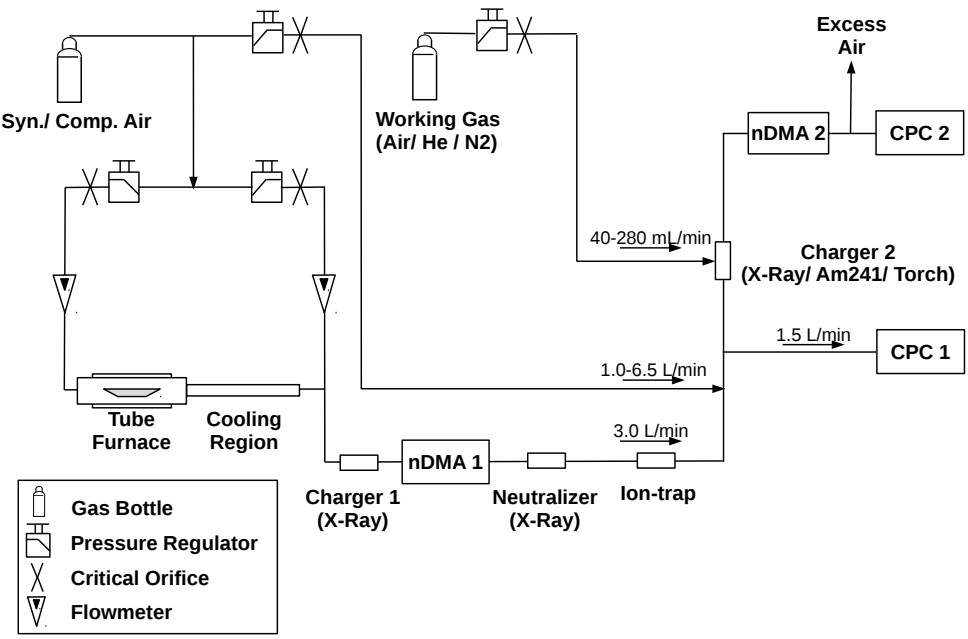

**Figure 1.** Schematic of the experimental setup for the charging probability and particle size conservation measurements. The additional working gas flow is only needed for the plasma charger. See text for explanation.

To secure that no charged particles remain in the aerosol flow an ion-trap was installed after the bipolar diffusion charger (Brilke et al., 2020). After the ion-trap the aerosol flow was split in a way that 1.5 L/Min were led to a CPC (TSI 3776 UCPC) which recorded the particle concentration at this point while the remaining aerosol flow was fed into the bipolar diffusion charger under investigation. A dilution flow of synthetic air as well as compressed air allows for varying the flowrate before the aerosol flow gets to the bipolar diffusion charger. The investigated bipolar diffusion chargers were switched in intervals of 10 minutes during the measurements to secure that the different devices were operated at comparable conditions. Afterwards the charged aerosol flow was led to a second nDMA, which was operated in scanning mode. A second CPC (TSI 3776 UCPC) recorded the particle concentration at this point of the setup. In addition to the switching of the different bipolar diffusion chargers, the working gas as well as the working gas flow of the atmospheric pressure plasma source were varied.

Complementary to the charging efficiency, the chemical composition of the charger ions were investigated by coupling the plasma torch with an APi-TOF MS (Atmospheric Pressure Interface - Time of Flight Mass Spectrometer, Junninen et al. (2010); Leiminger et al. (2019), see upper panel in Figure 2), to analyze the chemical composition of the ions generated in positive and negative ion mode. Mobility spectra were recorded with a custom-built Faraday Cup Electrometer (FCE) (Winklmayr et al., 1991) with an improved response time of 0.1 s. By recording the ion spectrum with a Vienna-type high-resolution mobility analyzer (UDMA-1, Steiner et al. (2010)), the mobility equivalent diameter of the generated clusters could be analysed (see

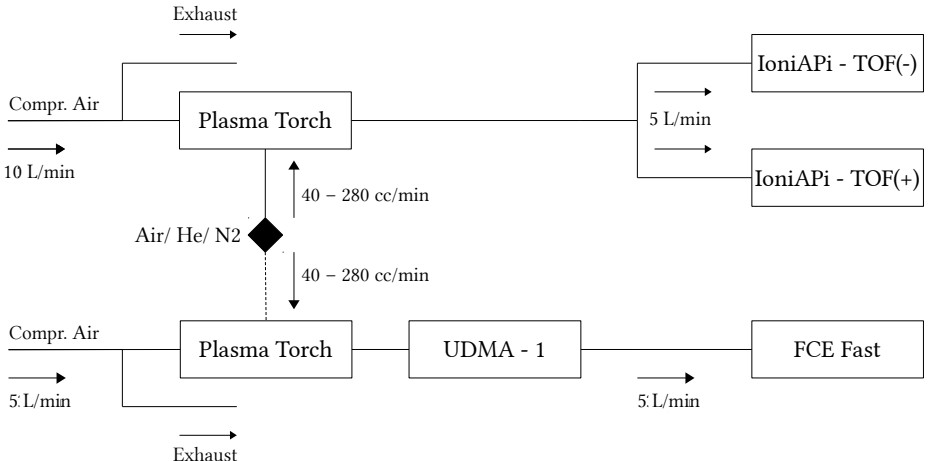

**Figure 2.** Schematic of the calibration setup for the plasma torch using an UDMA. The black rhombus marks the working gas supply. Mass spectra of negative and positive ions were measured simultaneously using the ioniAPi-TOF in negative and positive ion mode at a flow rate of 10 L/min through the plasma torch (upper panel) for the three working gases and operational settings. The mobility spectra of the generated positive and negative ions were measured using the UDMA-1 (Steiner et al., 2010) coupled to a fast-response FCE at 5 L/min flow rate.

lower panel in Figure 2). Compressed filtered and dry air was used as carrier gas and the relative humidity (RH) was monitored using SHT75 RH sensors with an accuracy of $\pm 1.8\%$ and was kept below $2\%$. However, we cannot exclude that in the closed loop sheath flow system of the UDMA-1 small amounts of water vapor remain. The calibration of the ioniAPi-TOF mass axis was performed using a bipolar electrospray source for the generation of tetra-heptyl ammonium bromide clusters (Fernández de la Mora and Barrios-Collado, 2017; Brilke et al., 2020). The UDMA-1 resolution power is 15 at the size of the THABr monomer, i.e. 1.45 nm mobility equivalent diameter (Flagan, 1999; Steiner et al., 2010). Due to the high ion concentration of a non-thermal plasma source a vast number of reactive species are created, especially when the plasma is ignited in air (Kurake et al., 2016). Most of these species will be ozone or nitrogen oxides because of the air's chemical composition. Hence, the optical emission spectra of the non-thermal plasma source were recorded, which is a non-invasive diagnostic technique that allows to gain insights into the composition of the plasma and the production of harmful gases like ozone and nitrogen oxides. The optical emission spectrometer was located at the nozzle of the plasma charger and used to record spatially averaged optical data along the axis of the plasma source.

## 3 Results and Discussion

### 3.1 Optical Emission Spectroscopy

Optical emission spectroscopy (OES) was used to determine the ionization stages of the ions/molecules in the plasma. The measurements were performed with the HR2000+ES spectrometer from Ocean Optics. The light emission from the atmospheric

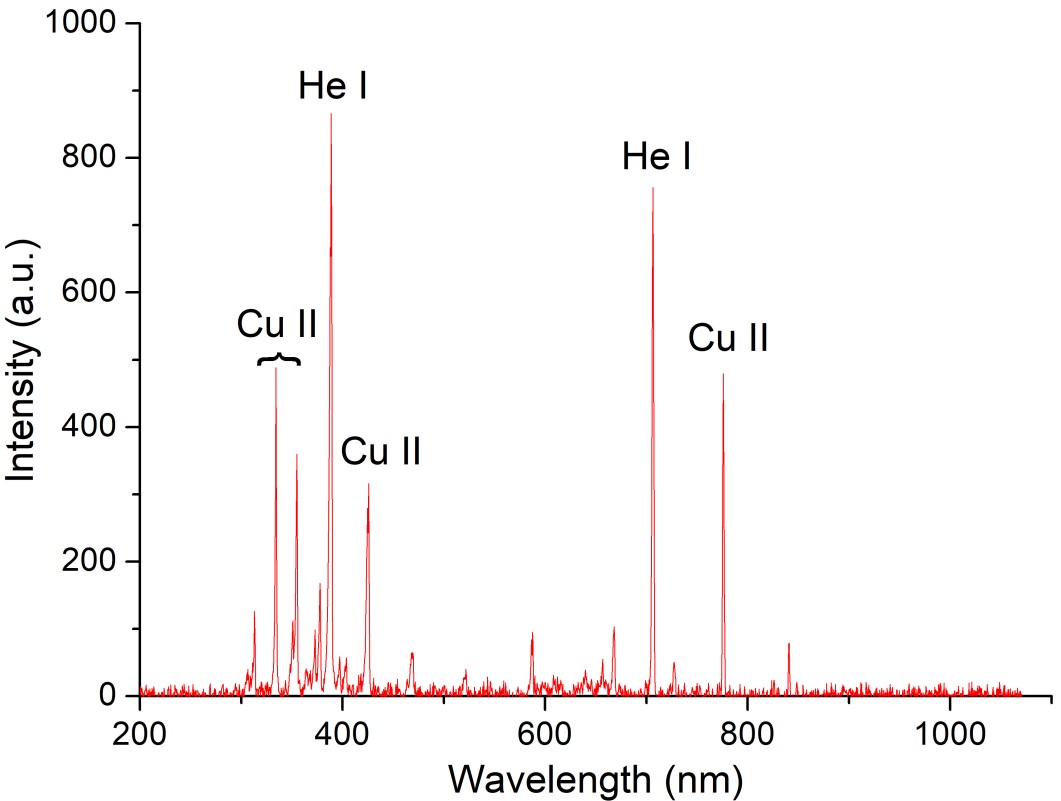

**Figure 3.** Typical OES spectrum in a helium plasma averaged over 50 scans. Experimental conditions: 15 kHz driving frequency, 825 volts peak-to-peak and 180 mL/Min He flow.

pressure plasma source was collected with a 440 $\mu m$ fiber with a length of 2 m. The wavelength range of the spectrometer was

between 200 and 1100 nm. Prior to the data acquisition the torch was switched on for about 30 seconds until no fluctuations in the spectra were visible. Each spectrum is the result of averaging over 50 scans to remove fluctuations. The plasma source itself was driven with a high frequent alternate current of 15 kHz with 825 V peak-to-peak voltage. The plasma was ignited in helium of high purity (ALPHAGAZ 1 HELIUM, $>=99.999\%$ (5.0), Air Liquide), which was fed into the plasma jet with a flow rate of 180 mL/min. The plasma is ignited inside the inner flow while the aerosol is administered through the outer gas

stream. The main source of the plasma is a high-frequency copper antenna/electrode that is situated on the central axis of those two gas streams. The results for a typical OES spectrum for the described experimental conditions are depicted in Figure 3.

The most prominent emission lines were identified to be excited neutral helium (He I) and singly ionized copper (Cu II). The central wavelengths of the identified lines are listed in Table 2.

It can be concluded from the OES spectra that there is no ionization of aerosol particles facilitated by the carrier gas, since

only neutral helium emission lines have been recorded. On the other hand, atoms from the copper high frequency antenna enter the plasma zone and are then ionized through electron impacts. The charged particles created from these processes (i.e. ions

**Table 2.** Measured Central wavelengths compared to data from Kramida et al. (2013) and the associated particle species for the most prominent emission lines in Figure 3.

| Measured central wavelength [nm] | Wavelength from Kramida et al. (2013) [nm] | Particle Species |
|---|---|---|
| 334.38 | 334.4 | Cu II |
| 354.86 | 354.9 | Cu II |
| 388.86 | 388.9 | He I |
| 425.67 | 425.6 | Cu II |
| 706.05 | 706.5 | He I |
| 755.56 | 775.4 | Cu II |

and secondary electrons), in turn, charge the aerosol particles in the gas stream. This can be explained by the large difference in ionization energies between copper (7.7 eV) and helium (24.6 eV). Since the electrons in non-thermal atmospheric pressure discharges have normally energies of just a few eV, the ionization of copper atoms is far more likely than of helium particles.

## 3.2  Charger ion chemical composition

The ion properties of ionic molecular clusters produced in the plasma torch were investigated by means of electrical mobility and mass spectrometry. Mobility spectra and mass spectra were recorded for positive and negative ions and compared to the resulting spectra from ions produced in the $^{241}$Am charger. Figure 4 shows the mass spectra for negative (left) and positive (right) ions generated by the $^{241}$Am charger (first panel) and the plasma torch for the three different working gases (second to fourth panel) using the setups shown in Figure 2 at the working gas flow settings presented in Table S1 and S2 in the supporting information (SI). The negative mass spectra were normalized to the nitrate ion ($NO_3^-$) peak at an integer mass of 62 Th and the positive mass spectra to the $(H_2O)_2 \cdot H_3O^+$ water cluster at an integer mass of 55 Th. The negative mass spectra are dominated by the nitrate ion, $NO_3^-$, and its dimer, trimer and water cluster (see labels in second panel in Figure 4). The three spectra for charger ions produced from the torch exhibit the same major peaks with the nitrate ion trimer peak being highest when He is used as working gas (see fourth panel). This observation may be a result of the different operational settings of the plasma torch when He is used as working gas (see Table S1 and S2). The negative ion spectrum of the $^{241}$Am charger reveals the same major peaks as the plasma torch negative mass spectra. Similar results have been found by a study investigating the chemical composition of ions produced by a corona discharge (Manninen et al., 2011). The identified major peaks of the positive mass spectra are listed in Table 3. In the lower mass range between 40 - 80 Th, protonated water, $H_3O^+$, and water clusters thereof dominate the spectrum for the $^{241}$Am charger and the three spectra of the plasma torch. The elemental composition of four major peaks in the positive spectra of the plasma torch at integer masses of 88, 175, 187 and 201 Th were identified as carbonaceous compounds. In the higher mass range between 600 - 1000 Th, a set of major peaks was identified as silicone compounds. The same peaks were identified in the positive mass spectra of ions produced from the $^{241}$Am charger (solid black box in Figure 4). This observation was also made by Manninen et al. (2011), who explain these peaks as a result of

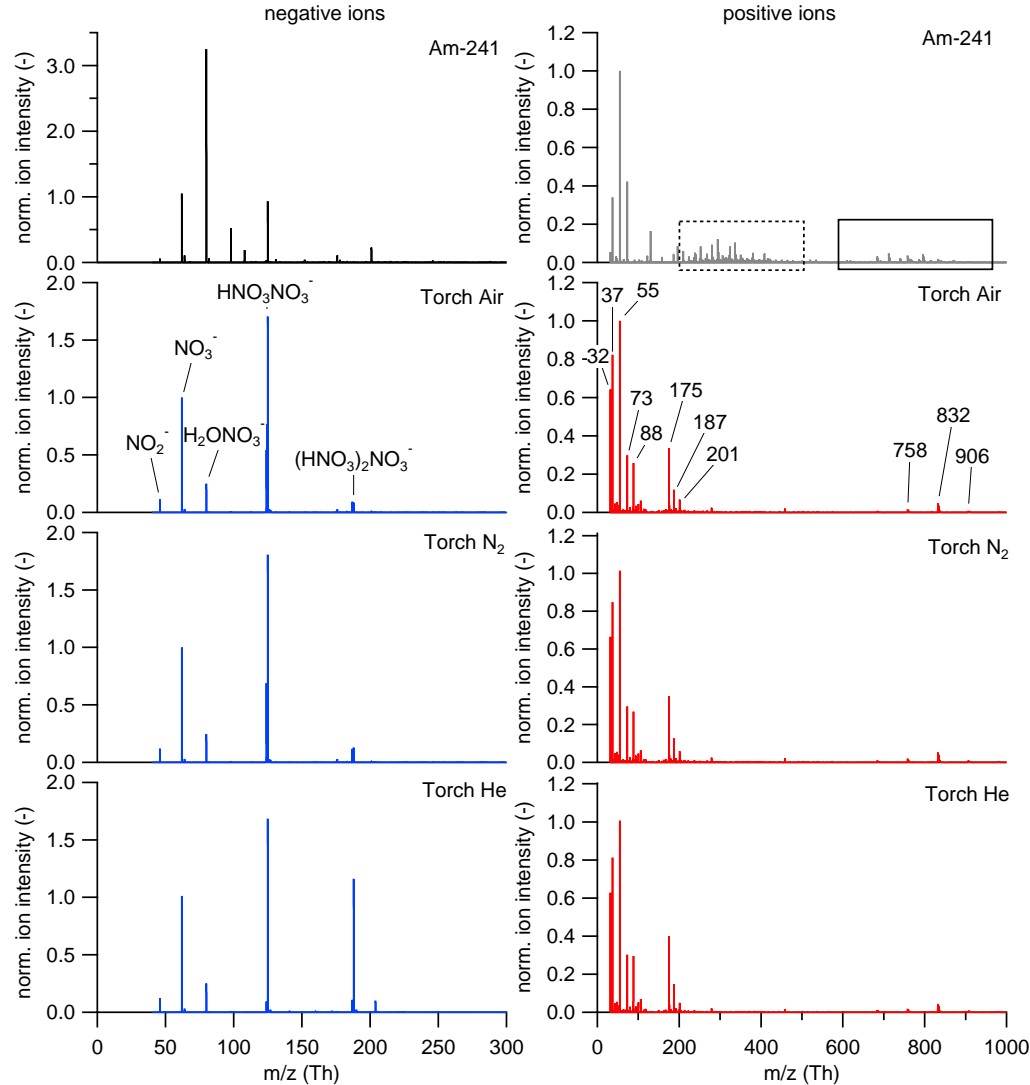

**Figure 4.** Negative (left) and positive (right) mass spectra of ions generated by the $^{241}$Am charger (first panel) and the plasma source for three different working gases, synthetic air (second panel), $N_2$ (third panel) and He (fourth panel) were measured using the setup in the upper panel of Figure 2. The mass spectra were averaged over 1 hour each. The identified compounds are labelled in the second panel and are presented in Table 3. The negative mass spectrum was normalized to the $NO_3^-$ ion (integer mass 62 Th), the positive mass spectrum was normalized to the $(H_2O)_2 \cdot H_3O^+$ cluster (integer mass 55 Th). The $H_3O^+$ ion is not displayed here since it was not covered by the set mass range of the ioniAPi-TOF. The dashed square box marks unidentified masses in the positive $^{241}$Am mass spectrum and the solid square box shows the silicone compounds that are listed in Table 3.

contamination from silicone tubing. Silicone tubing is oftentimes used in aerosol measurements and can cause artifacts because of degassing of siloxanes (Asbach et al., 2016). However, a range of peaks in the mass window from approximately 200 to 500

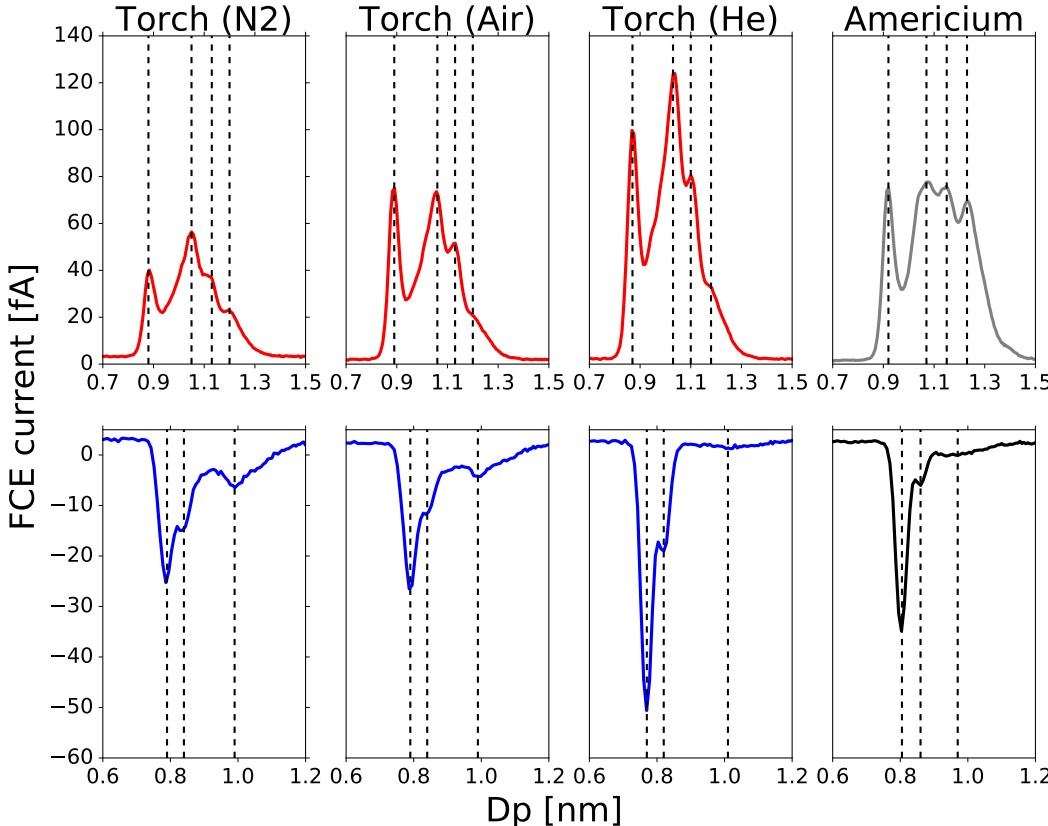

**Figure 5.** Mobility distributions of the charger ions generated by the plasma torch using $N_2$, He and air as working gas and the $^{241}$Am charger. The upper panel shows the mobility diameter distribution of the positive charger ions, the lower panel presents the mobility diameter distribution of the negative charger ions.

Th remains unidentified (dashed black box). These carbonaceous compounds have a positive mass defect and likely arise from ionization of constituents in the pressurized air that was used as carrier gas.

The chemical composition of the plasma-generated ions was found to be independent of the choice of the working gas as shown in Figure 4. Also, the averaged electrical mobility measurements (averaged over 10 scans) conducted with the experimental setup depicted in Figure 2 (lower panel) revealed identical peaks. The mobility spectra for the plasma torch using $N_2$, He and air as working gas and the $^{241}$Am charger are presented in Figure 5. Similar results for different bipolar charging devices have been found by Kallinger et al. (2012) and Steiner and Reischl (2012). The latter have analysed the effects of carrier gas contaminants on the charging probability which influences the electrical mobility spectrum. One of the analysed TSI X-Ray charger showed a different mobility spectrum compared to the other analysed bipolar diffusion chargers. According to our mass spec analysis, this is due to ammonium sulfate contaminations from previous experiments.

**Table 3.** Overview of major negative and positive compounds in the mass spectra recorded using the ioniAPi-TOF in positive and negative ion mode.

| Negative Ions | | Positive Ions | |
|---|---|---|---|
| Integer m/z (Th) | Molecular Formula | Integer m/z (Th) | Molecular Formula |
| 46 | $NO_2^-$ | 32 | $O_2+$ |
| 62 | $NO_3^-$ | 37 | $H_2O \cdot H_3O^+$ |
| 80 | $H_2O \cdot NO_3^-$ | 55 | $(H_2O)_2 \cdot H_3O^+$ |
| 124 | - | 73 | $(H_2O)_3 \cdot H_3O^+$ |
| 125 | $HNO_3 \cdot NO_3^-$ | 88 | $C_4H_{10}NO^+$ |
| 187 | - | 175 | $C_4H_9NO \cdot C_4H_{10}NO^+$ |
| 188 | $(HNO_3)_2 \cdot NO_3^-$ | 187 | $C_{10}H_{21}NO_2^+$ |
| | | 201 | $C_{11}H_{23}NO_2^+$ |
| | | 610 | $(SiOC_2H_6)_8OH_2^+$ |
| | | 684 | $(SiOC_2H_6)_9OH_2^+$ |
| | | 758 | $(SiOC_2H_6)_{10}OH_2^+$ |
| | | 832 | $(SiOC_2H_6)_{11}OH_2^+$ |
| | | 906 | $(SiOC_2H_6)_{12}OH_2^+$ |

## 3.3 Charging Probability

The bipolar diffusion chargers were tested in the setup shown in Figure 1, in order to characterize the charging performance of the plasma torch (Gilbert Mark I plasma charger) and the [241]Am as well as the soft X-Ray charger. The tandem DMA setup enabled us to charge a monodisperse aerosol and rotate the different bipolar diffusion chargers, which permits a direct comparison of the charging performance of the different devices. Two butanol-based CPCs (TSI 3776 UCPC) with reduced temperature settings (Condenser 1.1°C, Saturator 30.1°C, Optics 31.1°C) compared to factory settings to increase the particle counting efficiency were used (Barmpounis et al., 2018; Tauber et al., 2019a). The particle number concentration was recorded before (CPC1, See Fig. 1) and after charging (CPC2) by the tested charger (Charger 2). The charging efficiency was inferred from the ratio of the two CPCs (CPC2/CPC1) under consideration of the transmission and diffusional particle losses in the lines and DMA. In Tauber et al. (2019b) the particle counting efficiency of the CPCs used here was determined, and the results obtained were used to correct for the CPC detection efficiency. The different bipolar diffusion chargers were tested with positively and negatively charged Ag and NaCl particles of different particle sizes in the sub-12 nm regime. In addition, the plasma torch was operated with different working gases. The additional flow rate from the working gas was at max 1/9 of the aerosol flow. According to Thomas et al. (2018) a cutoff drift to lower sizes for helium mole fractions below 0.67 was found for butanol-based CPCs. However, the used CPC in this study was operated with reduced temperature settings and thereby a lower detection efficiency was established. As a result, the recorded cutoff drift would therefore only influence the charging

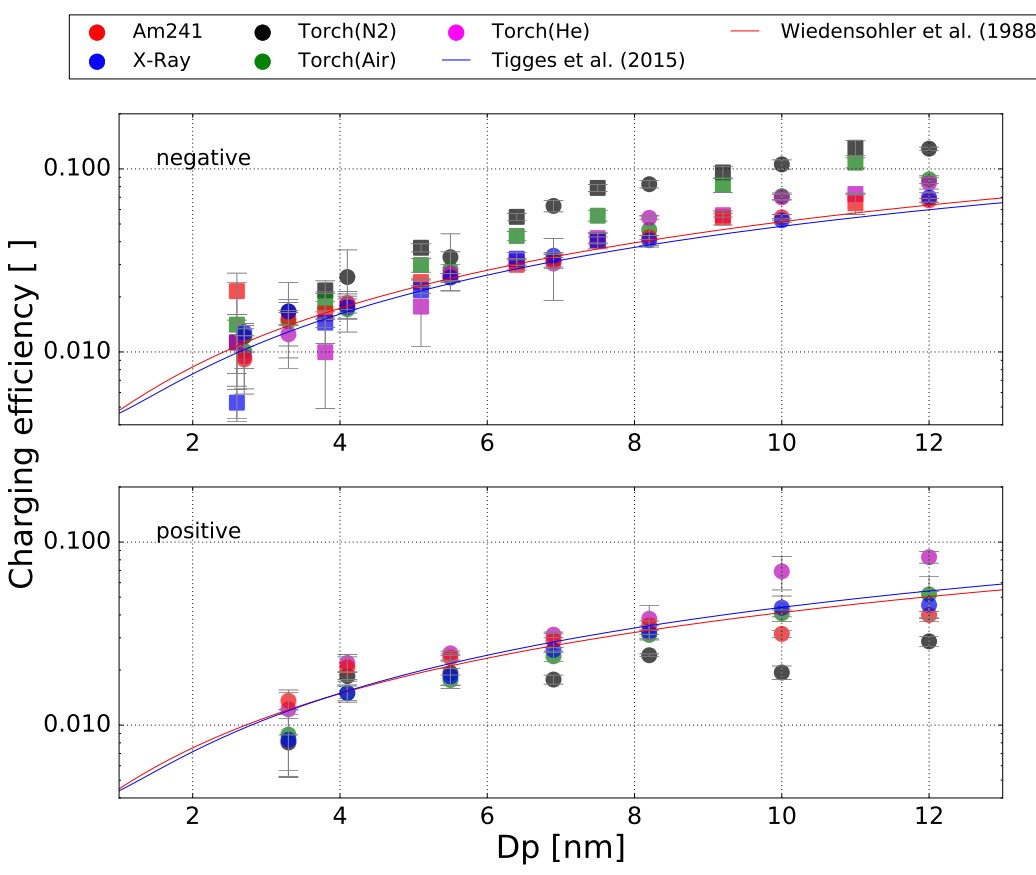

**Figure 6.** Measured charging efficiencies for the different aerosol chargers for negatively and positively charged Ag particles (dots) and negatively charged NaCl particles (squares) with mobility diameters less than 12 nm.

efficiency measurements conducted at < 3 nm. The resulting error is already covered for this particle sizes by the measurement uncertainties of the nDMA and CPC.

The results of the charging efficiency when using the three different working gases are displayed in Figure 6. Theoretical charging probabilities from Tigges et al. (2015) and Wiedensohler (1988) were added here.

    Concerning negatively charged particles with a diameter between 4-12 nm the [241]Am and the soft X-Ray charger are in agreement with the theoretical curves especially with Wiedensohler (1988). The plasma torch on the other hand achieves higher charging efficiencies in this regime with big differences between the used working gases. With helium the charging

efficiencies were higher than the theoretical values, especially for particle sizes bigger than 8 nm a 25 % higher charging efficiency was recorded. Ambient air yields even higher charging probabilities but displays a significant dependency on the particle type as the values for NaCl particles are about 25 % higher than for Ag particles. The highest charging efficiencies for negatively charged particles were achieved with nitrogen as working gas. With nitrogen the plasma torch achieved charging efficiencies that were up to 50 % higher than the common devices and showed no dependency on the particle type. As the

particle concentrations for very small diameters between 2-4 nm are below 10000 #/ccm, the charging efficiencies of the different devices vary strongly at these particle sizes. The nDMA in this size range has a low transmission efficiency ranging from 20-55 % and therefore the signal at the CPC 2 is very low. In addition, small temperature-fluctuations in the tube furnace lead to bigger uncertainties for the low total number concentration of the selected particle size. Especially for NaCl particles a big variation in the charging efficiencies of the different charging devices was observed. Those variations almost certainly are caused by the low number concentrations of NaCl particles at these sizes compared to Ag particles. However, except for NaCl particles at particle diameters between 2-4 nm the associated data points for the [241]Am bipolar diffusion charger agree well with the approximations by Tigges et al. (2015) and Wiedensohler (1988).

The charging efficiencies of the plasma torch for positively charged particles strongly differ from those of negatively charged particles. Again, the measured charging efficiencies of the soft X-Ray charger match almost perfectly with the theory for diameters between 4-12 nm. The [241]Am charger also agrees with the theoretically predicted values for diameters between 6-8 nm but shows asymmetries for larger particle sizes. The charging efficiency for positively charged particles decreases at larger sizes and results in lower values compared to theory for the [241]Am charger. The charging efficiencies of the plasma torch strongly depend on the type of working gas for positively charged particles. Evidently, the data using helium agree well with theoretical approximations from Tigges et al. (2015) and Wiedensohler (1988) in the size regime between 4 and 10 nm. For diameters between 10-12 nm the charging efficiencies even exceed the predicted values by about 50 %. The data suggest no dependency of the charging efficiency on the charger ion polarity, when helium is used as working gas in the plasma torch. The data for compressed air match with the theoretical curves for the whole size range, and therefore behave similar to the soft X-Ray charger. As Figure 6 clearly shows, this will change drastically if nitrogen is used as the working gas. The charging efficiencies in positive polarity with nitrogen are significantly lower (about 50 % compared to theory) than with other chargers and other working gases in the whole size regime.

Wiedensohler and Fissan (1991) have shown that the predicted charging probabilities of NaCl and Ag particles strongly depend on the used carrier gas and the ion mass and mobility. During the ionization process positive ions and free electrons are formed from molecules in the carrier gas and ionized copper atoms. These primary ions attach to other molecules, as for example $H_2O$, $CO_2$, oxygen and halogens, and form bigger ion clusters that afterwards stick to the investigated particles. Wiedensohler and Fissan (1991) have shown that the variation of the ion masses leads to different theoretically predicted charging probabilities. For nitrogen as carrier gas they discovered a large dependency of the charging probability on the ion masses. Similar asymmetries are observed in our data when comparing the measured charging efficiencies of the plasma torch with nitrogen as working gas to the $N_2$ results of Wiedensohler and Fissan (1991). As Figure 3 shows, the plasma torch forms copper ions and free electrons which charge aerosol particles in the carrier gas. The significantly different masses of these ions may account for the differing charging efficiencies that are accomplished with nitrogen for the negatively as well as for the positively charged particles according to Wiedensohler (1988).

As the working gas flow is exposed to a high frequency electrical field before it mixes with the aerosol flow, the ions can form in a pure nitrogen environment. Hence the charger ions form in a nitrogen atmosphere like in the case of Wiedensohler and Fissan (1991) where silver and sodium chloride particles were charged with a Kr 85 source in a pure nitrogen atmosphere.

**Table 4.** Comparison of ion cluster properties: polarity, mobility diameter $D_p$ calculated from mean ion mobility $Z$, mean ion mass $M$ and ion mobility ratio $Z^-/Z^+$.

| | Polarity | $D_p$ [nm] | $Z$ $[cm^2/Vs]$ | $M$ [Da] | $Z^-/Z^+$ | |
|---|---|---|---|---|---|---|
| Reischl et al. (1996) | + | 1.32 | 1.15 | 290 | 1.0 | 0.80 |
| Reischl et al. (1996) | - | 1.19 | 1.43 | 140 | 1.0 | 0.80 |
| measured | + | 1.07 | 1.76 | 356 | 1.0 | 0.66 |
| measured | - | 0.87 | 2.66 | 116 | 1.0 | 0.66 |

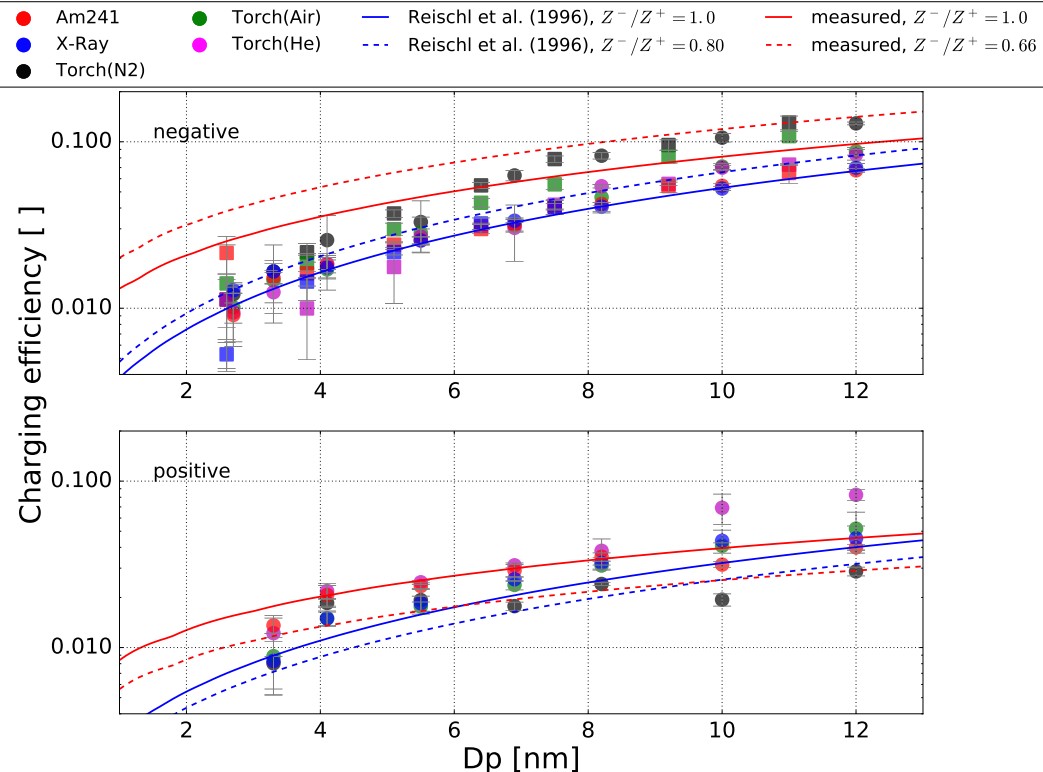

**Figure 7.** Measured charging efficiencies for the different aerosol chargers for negatively and positively charged Ag particles (dots) and negatively charged NaCl particles (squares) with mobility diameters less than 12 nm. The lines represent the charge distribution according to Fuchs theory, the parameters for the ion mobilities, ion masses and ion mobility ratio are listed in Table 4. The collision probability of ions was calculated following Hoppel and Frick (1986).

A similar but smaller effect was observed in atmospheric air as carrier gas. This mechanism would also explain the better charging efficiencies with ambient air as working gas.

In Table 4 the measured and calculated mean ion charger mobilities, mobility equivalent diameters, masses and ion mobility ratios are listed. For comparison the values found by Reischl et al. (1996) are also listed. The results were used to calculate the

charge distribution with Fuchs theory as shown in Figure 7. Negatively charged particles in the size range from 4-12 nm by [241]Am or X-Ray bipolar diffusion chargers agree well with the parameters derived by Reischl et al. (1996) and an ion mobility ratio of 1. For positively charged particles the charging efficiency is below the measurement results for particles between 4-10 nm. By correcting the charge distribution with the parameters derived by Reischl et al. (1996) with an ion mobility ratio of 0.8 the negatively charged particles with a size below 4 nm fit perfectly to theory. The measurement results of this work reveal an increased charging efficiency for both polarities as shown in Figure 7. For mobility equivalent diameters between 4 and 12 nm and positive polarity the charging efficiency fits with theory for [241]Am, X-Ray and the plasma torch with air as working gas. In contrast to negatively charged particles where the results of the plasma torch with nitrogen or air as working gas above 7 nm is higher and below 7 nm is lower than expected by theory. Also, for [241]Am, X-Ray and the plasma charger with helium the theory exceeds the measured charging efficiency. By correcting the theory with the acquired ion mobility ratio, a good agreement between theory and measurement can be found for negatively charged particles above 7 nm and for positively charged particles for the plasma torch with nitrogen as working gas. Although the effect for diameters > 7 nm can be explained, there is still a deviation for the smaller diameters from theory. The reported discrepancy can therefore not solely be attributed to the ion mobilities. There are other effects which should by investigated in further studies. Especially the charging effects below 5 nm which cause deviations from the charging model.

Figure 8 depicts the recorded negatively charged particle number size distribution averaged over numerous measurements. The inversion of the size distribution data was performed according to Petters (2018). These diagrams permit a qualitative descriptive comparison of the different charging devices. The plot reveals a shift of about 1.3 nm in the maximum of the recorded size distribution between the X-Ray and the plasma charger. In this study we assume the resulting deviation in size distribution is two-fold. Firstly, contaminations of the X-Ray charger from previous experiments with ammonium sulfate lead to an increase of about 0.7 nm in particle diameter for the observed bipolar diffusion charger (Steiner and Reischl, 2012). Secondly, the application of a unsuitable charge distribution for the plasma charger leads to a decrease of the measured particle diameter of about 0.5 nm. By applying the measured charge distribution for the plasma torch with $N_2$ as working gas prior to data inversion the deviation in size vanishes and it is comparable in particle diameter with the measured size distribution of the [241]Am bipolar diffusion charger (see Figure 8 (Torch (N2)*)).

In addition, a significantly higher peak (about 50 %) with the plasma torch was recorded which is most likely due to the higher charging efficiency. Since the bipolar diffusion chargers were rotated periodically in multiple cycles, the possibility of systematic uncertainties in the actual size distribution was minimized. Therefore, the raised signal of the plasma torch can be attributed to a generally higher charging efficiency. The X-Ray charger is slightly less efficient than the radioactive [241]Am charger but still within the uncertainty range. This decrease can be attributed to the performance reduction during continuous operation which typically occurs during long measurement cycles. Compared to conventional chargers the plasma torch proves to charge slightly better with air as the working gas and less efficient with helium. With nitrogen as working gas, the plasma torch charges up to 50 % more than with air and helium and more than the other tested bipolar diffusion chargers. According to Maißer et al. (2015), nitric acid has an anomalously high gas phase acidity for its mass and can persist in the gas phase in higher concentrations than other low mass species. By using helium as working gas the concentration of nitrate ions in the gas

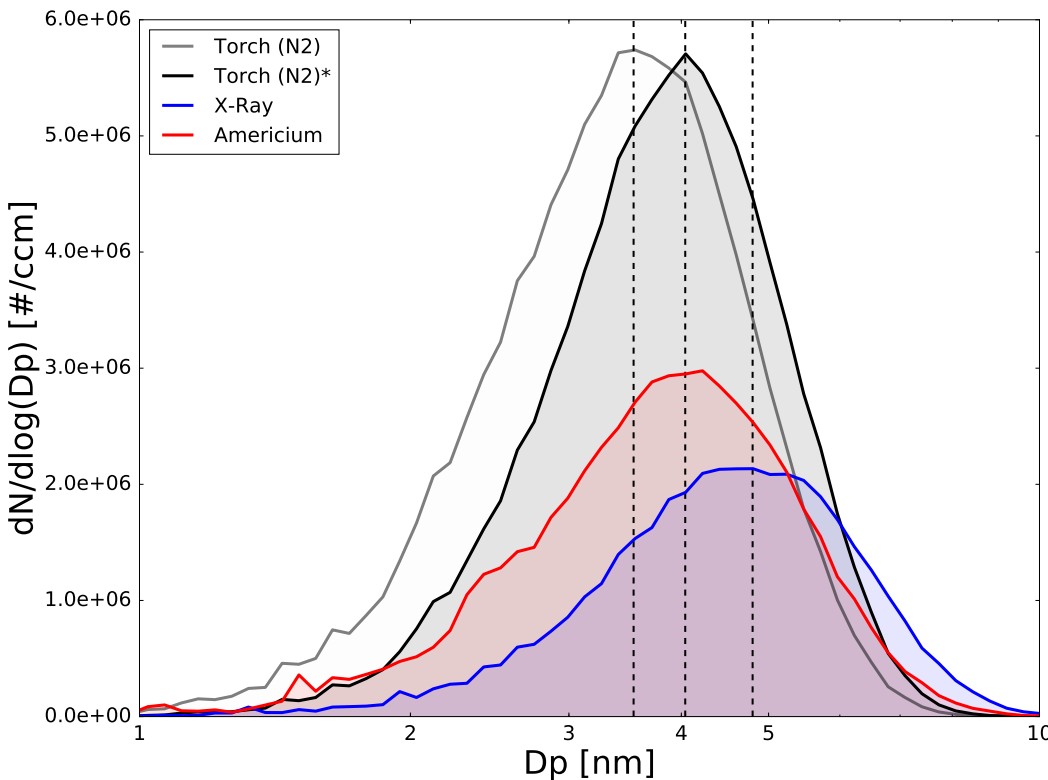

**Figure 8.** The plot represents the same recorded negative Ag particle number size distribution with a UCPC, depending on the mobility diameter for the plasma torch (N2), X-Ray and americium bipolar diffusion chargers. Furthermore, the shift of the size distribution caused by the application of a unsuitable charging efficiency (Torch (N2)) for the plasma torch and the measured charging efficiency (Torch (N2)*) and due to contaminations of the X-Ray charger are shown.

phase is lower than in air or $N_2$ and therefore charge transport decreases. This is contrary to using $N_2$ as working gas where
an increased charging efficiency up to 50% was measured.

Figure 9 shows the dependence of the charging efficiency for different aerosol flowrates and particle sizes. The charging efficiencies for particles in the sub-50 nm regime show a significant dependence on the aerosol flowrate. The different bipolar diffusion chargers have proven to be more sensitive to varying flow rates and a reduced residence time in the ionizing atmosphere (He and Dhaniyala, 2014; Kallinger et al., 2012; Kallinger and Szymanski, 2015).

Figure 9 reveals that the aerosol charging is most efficient for a flowrate of 2.5 L/Min and the charging efficiency decreases for higher flowrates. Kallinger and Szymanski (2015) and Jiang et al. (2014) also measured the flowrate dependence of different chargers. In the study by Kallinger and Szymanski (2015) an increased charging efficiency for the americium bipolar diffusion charger for a flowrate of 5.0 L/Min was found. In addition, contradictory to our findings, a reduced charging performance for the X-Ray bipolar diffusion charger was not observed by Kallinger and Szymanski (2015) and Jiang et al. (2014). A reason
for that could be a reduced charging due to low power output of the X-Ray tube. Since the last repair and calibration of the

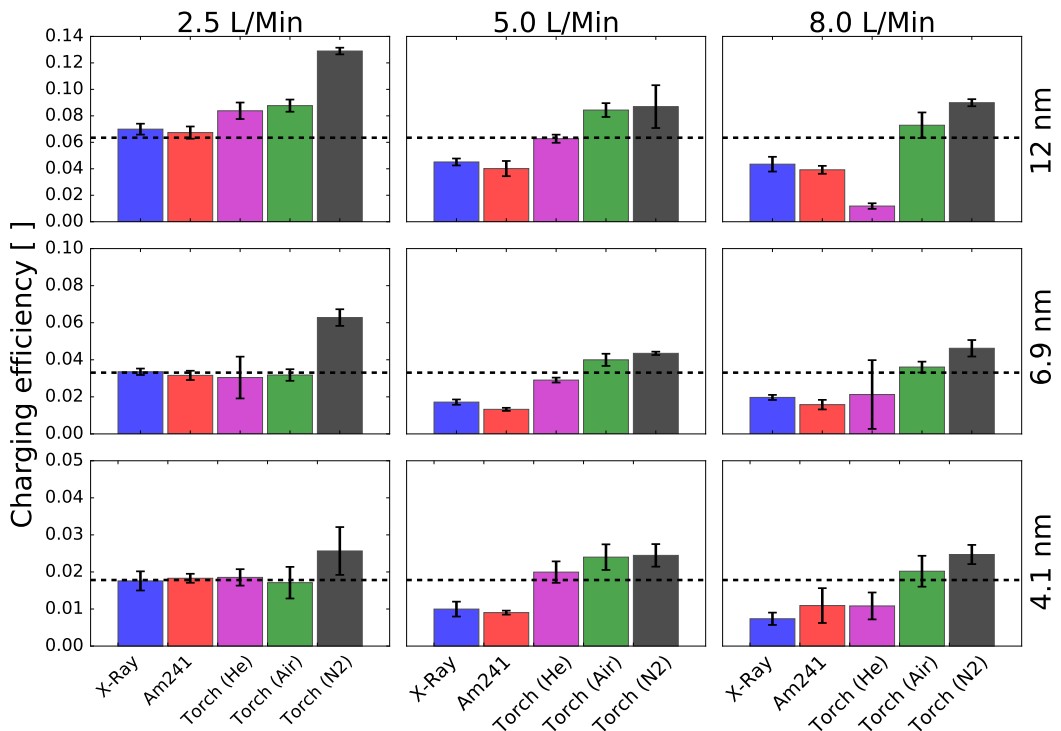

**Figure 9.** Measured charging efficiencies for the different aerosol chargers and aerosol flow rates for negatively charged Ag particles with mobility diameters of 12, 6.9 and 4.1 nm. The dotted black line represents the theoretical charging efficiency according to Wiedensohler (1988) for the 3 different mobility diameters.

analyzed TSI 3088 neutralizer the operating run-time was about 371 hours. Furthermore, in this work, we used a different aerosol generation method compared to the above mentioned studies which could lead to different charging mechanism. Especially in the sub-10 nm size range different chemo-physical interactions might lead to unforeseen results which should be investigated in future studies. The plasma torch also achieves the highest charging efficiencies for lower flowrates but seems to be not as sensitive to the aerosol flow compared to the other devices. Especially air and nitrogen have proven to be the most robust options as working gases.

In addition to the flow dependent charging efficiency measurements, the ozone concentration was recorded with an O3-Monitor (Thermo Scientific Fischer, Model 49i). Elevated ozone concentration were observed at lower flow rates and for air as well as helium as working gas. In the case of air, a high amount of oxygen is present in the working gas of the plasma torch which supports the ozone production. For helium the afterglow of the plasma torch and the corresponding UV-light emission can be responsible for the increased amount of ozone. In addition, the applied electromagnetic field can cause a split of $O_2$ molecules. Especially, when the residence time is increased at low working gas flow rates. All recorded measurements can be found in the supporting material (Table S1).

## 4 Conclusions

The presented measurements conducted with a non-thermal plasma source have shown that helium, nitrogen and air as working gases lead to the same ion species. According to the mobility and mass spec measurements the comparison of the plasma charger with the americium bipolar diffusion charger indicates the same negative ion species. Whereas for the positive ions the measurements reveal a slight deviation. At this point it should be noted that the chemical composition of the charger ions is affected by the tubing material and contaminations as discussed by Steiner and Reischl (2012).

The analyzed chemical composition of the bipolar diffusion charger ions did not lead to changes in chemical composition even with increased ozone concentration caused by the plasma torch. By switching the working gas to nitrogen an increased charging efficiency could be recorded for negatively charged particles compared to [241]Am bipolar diffusion charger. In accordance with Mathon et al. (2017), the ozone concentration can be reduced to ambient conditions with nitrogen which is beneficial for the commercial use. For future studies, the influence of the variation of the operational settings and the resulting ozone concentrations of the plasma torch remains to be investigated. The option of setting high and low ozone concentrations are of interest when analyzing the chemistry of different particles, for example proteins (Kotiaho et al., 2000). In addition, the results would also be useful for the application of corona discargers, which produce ozone as well.

The higher charging probabilities for negatively charged particles in the size range from 7 to 12 nm can be attributed to differences in the electrical mobility. According to our measurements the ratio of ion mobilities, given by the ion mobility of positively charged particles divided by the ion mobilities of negatively charged particles, yields a constant value of 0.66. Consequently, the ion mobilities for negatively charged particles are on average higher. This increased charging efficiency for negatively charged particles and decreased charging efficiency for positively charged particles was also measured by Wiedensohler and Fissan (1991) for Kr85 bipolar bipolar diffusion charger in nitrogen. According to our measurements, similar results could be found for the plasma torch with nitrogen as working gas. However, our results reveal a strong deviation to classical charging theory in the observed size range which has to be considered prior to data inversion in future laboratory or field applications.

The charging efficiency of the non-thermal atmospheric plasma source indicated a weak aerosol flow dependence when operated with nitrogen or compressed air in comparison to the americium and X-Ray bipolar diffusion chargers. As a result, the application of the plasma charger for increased flow rates in laboratory applications is promising.

In summary, with different experimental approaches we were able to quantitatively characterize the Gilbert Mark I plasma source with nitrogen, helium and air as working gas. In addition, a commonly used X-Ray bipolar diffusion charger and a radioactive americium bipolar diffusion charger were analyzed for comparison. The highest charging efficiencies for negatively charged particles were found for the Gilbert Mark I plasma charger with nitrogen as working gas. Our results also reveal the importance of well characterized and clean bipolar diffusion chargers to avoid any misinterpretation of experimental data especially in the sub-12 nm size range. In addition, contrary to ozone suppression, the plasma source revealed great potential to act as an ozone generator by changing the working gas, for example, to argon.

*Data availability.* Supplementary data associated with this article can be found in the online version.

*Author contributions.* Christian Tauber designed the setup, Christian Tauber and David Schmoll performed the charging efficiency exper-
iments, Johannes Gruenwald and David Schmoll performed the OES measurements, Christian Tauber and Sophia Brilke performed the
325 mobility distribution measurements, Sophia Brilke and Daniela Wimmer performed the chemical composition measurements, Christian
Tauber, David Schmoll, Johannes Gruenwald, Sophia Brilke, Peter Josef Wlasits, Daniela Wimmer and Paul Martin Winkler were involved
in the scientific interpretation and discussion, and Christian Tauber, David Schmoll, Johannes Gruenwald, Sophia Brilke, Peter Josef Wlasits,
Daniela Wimmer and Paul Martin Winkler wrote the manuscript.

*Competing interests.* The authors declare that they have no conflict of interest.

*Acknowledgements.* This work was supported by the Austrian Research Promotion Agency (FFG) under grant number 870121 and by the
Austrian Science Fund (FWF) Project J3951-N36. The authors want to thank Peter Kallinger, Gruenwald Laboratories GmbH and Grimm
Aerosol Technik Ainring GmbH & Co Kg for the support.

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
