# Peer review of "Characterization of a Non-Thermal Plasma Source for the Use as a Mass Spec Calibration Tool and Non-Radioactive Aerosol Charger"

_Atmospheric Measurement Techniques, 2020_

## Referee Comment (RC1) · Anonymous Referee #1 · 24 Apr 2020

Manuscript by Tauber et al. reports characterization of a new bipolar non-thermal plasma type charger for charging sub-15 nm aerosol particles. The main results of the study are measured size dependent charging efficiencies that are strongly biased to negative polarity, and mass and mobility characterization of the charger ions. Experimental data on sub-10 nm charging efficiencies are in the general interest in aerosol measurement techniques, while the current manuscript requires elaboration on several places prior to consideration for publication. General and specific comments are listed below

General:

- Experimental section: there is absolutely no description of the new plasma charger which is being characterized. This undermines the reproducibility of the most important results of the whole manuscript. The operation principle and general characteristics of the charger have to be described prior to publication. Otherwise a serious scientist in another lab cannot reproduce the potentially interesting results obtained with the charger. This applies also to the other used chargers.

- P8 L136 onward and P15 L223 onward, the authors discuss the effect of ion mass on the charging probability. They have even measured the ion masses and mobilities, while still in the theoretical prediction they use presumably "wrong" ion masses and mobilities. Why? Why not predicting the charging efficiencies with the measured ion mobility and mass? This is also speculated in the conclusions with statements that the charging efficiencies biased to negative polarity are explainable with the ion mobilities, while not a single ion mobility value (which should be actually weighed with the ion concentration) is reported even if they are measured.

- Generally there are several experimental details that are not reported in sufficient detail, or statements of which interpretation is ambiguous. Some of these are picked below.

Specific:

- P1 L5, "increased charging efficiency", compared to what?

- P1 L7, "charging mechanism", are you referring to the different neutralizers? Charging mechanism is a different thing

- P1 L8, in which sense the TSI X-ray is "standard" neutralizer? I would say commercial

- P1 L9, "enhance" compared to what?

- P1 L10, "increased down towards", reformulate

- P1 L20, "below 10 nm in diameter are typically difficult to neutralize", I would say the

opposite

- P1 L21, "Quantitative particle detection in this size range is extremely challenging due to high diffusional losses. Hence, a higher charging efficiency is of importance to improve the signal intensity in the sub-10 nm regime", the first sentence implies challenges in determining the diffusion losses, i.e. systematic uncertainty, while the second one deals with detection limit and poisson type random uncertainty. Improvements in the second does not help the first one.

- P2 L33, it will be good to define what is non-thermal plasma, if it is the operation principle of the new charger

- P2 L39, "mobility spectrum of . . .", it is the mobility spectrum of the charger ions, not of the chargers.

- P2 L46, "steady-state", earlier you talked about equilibrium, which one it is?

- Figure 1, what is the sheath the flow rate and size resolution of the nDMAs? What about the furnace output distribution? How much the size resolution of the first DMA affects the results? Based on the plot, the DMA2 was operated with aerosol flow of 8 lpm at some point, can it really handle such high flow? Is it so that only the Torch requires the working gas? It is not clear from the figure. If the torch mixes helium with the sample flow, it changes the gas composition. Does it affect sizing of the DMA or the flow rates of the CPC? If yes, how was this accounted for?

- P2 L56, which kind of ion trap?

- P5 L93-94, "It can be concluded from the OES spectra that there is no ionization of aerosol particles facilitated by the carrier gas, since only neutral helium emission lines have been recorded." It is not clear what is the experimental setup during this experiment. It is only mentioned that helium is fed to the charger, there should be no particles in helium in the first place?

- P5 L94, "atoms from the copper high frequency antenna", what is this antenna?

- Section 2.1.1, what is the purpose of this section, or what we should conclude in terms of aerosol charging? Are the emission spectra comparable to the conditions of the charging efficiency experiments, or to the recorded mass spectra?

- P6 L103, "reduced temperature settings", what were the settings?

- P6 L107 "In Tauber et al. (2019b) the particle counting efficiency of the CPCs used here was determined, and the results obtained were corrected for the CPC detection efficiency", it is not clear what are used settings, since in the reference there are three different settings for the CPC.

- P6 L100-110, the description of the experiment should be in section 2 where the setup is described. How what about losses in the dilution, how they were corrected? The second DMA scans the voltage, what is the CPC2 signal that is compared to the CPC1?

- P6 L120, what the particle concentration has to do with charging efficiency? At extremely high concentrations concentration may affect the charging efficiency but not at low concentrations. Which CPC records the concentration that is mentioned? It is stated that data between 2-4 nm is in agreement with theory. Then the next sentence states this is not true for NaCl particles. Which one is true? Please elaborate the whole paragraph

- P7 L129, agrees with what?

- P7 L131, what is dependent on what?

- P8 L137 onward, also charger ion mobility should affect the charging probability.

- P8 L143 "As Figure 3 shows, the plasma torch forms copper ions and free electrons which charge aerosol particles in the carrier gas." Why these are not observed in the mass spectrum?

- P8 L147, "the ions can form in a pure nitrogen environment almost like in the mentioned case of Wiedensohler and Fissan (1991)", what do you mean?

- Figure 5. It is not described how the normalization is done for the right hand side plot. It is hard to believe that the differences in the distributions are only due to charging. At 8-10 nm the charging efficiency of the X-ray is about 10 times larger than of the torch which is opposite to what is reported in Fig4. On the other hand, at 3 nm the torch shows charging efficiency of a factor of 5 higher than the X-ray, which is also contradictory to fig4.

- Figure 6, the intro discusses with one sentence the nT product. Is that product sufficient in the chargers to reach the steady state charge distribution?

- P10 L185, ". The positive mass spectra were normalized to the nitrate ion ($NO_3^-$) peak at an integer mass of 62 Th and the negative mass spectra to the $(H_2O)_2$ ǔ $H_3O^+$ water cluster at an integer mass of 55 Th", the polarities do not match

- Section 2.1.3, previously there was discussion on the ionization of copper, why there is no copper signal in the charger ions?

- P12 L211, "$(NH_4)_{14}SO_4^-$", is it really a cluster of 14 $NH_4$ ammonium molecules?

- P15 L214, I would not say the are the same. Especially in positive there seem to be differences

- P15 L218, it was shown nowhere that the torch produces ozone.

- P15 L219, increased compared to what?

---

## Referee Comment (RC2) · Anonymous Referee #2 · 13 May 2020

This work describes the use of a plasma torch as an aerosol neutralizer. The work measures and compares the charging probability of the new source with commercial available other aerosol chargers. The charging probabilities were measured for positive and negative particles (Ag, and NaCl) and at different aerosol flow rates. In addition, the plasma torch has been evaluated at operation with different working fluids (He, Ar, N2). The mobility distributions and mass graphs of the charger ions were also measured in order to get an information on the properties of the charger ions. The work includes a very thorough investigation of the charger source and the charger ions. However, there are a few general concerns about the work, that should be addressed before publication:

1. The plasma torch itself, is not described at all in the current manuscript, a paragraph on the working principle should be added to the manuscript.

2. In the measurement with different gases, one would expect that using a DMA and CPC in a helium-air mixture would result into changes in the instrument performance. The voltage mobility relationship in the DMA is gas dependent, and in the CPC the flow calibration would change when adding helium to the system, also the supersaturation profile would change and the detection limit would shift to smaller particles (e.g. Thomas et al. 2018, Journal of Heat and Mass Transfer). These are points that should be addressed in this manuscript.

3. The charger ion mass and mobilities were measured, however, no qualitative thoughts were included in how and why that would result into the observed changes of charging probability. It has been stated correctly that the charger ion composition plays an important role to the final charge distribution. But what is missing is to apply the information found in this work to existing theories and see if the trend agrees with the observations. Simulation results of charge distributions considering different ion mass and mobility of charger ions have been performed in the past, see for example Maisser et al, 2015, Journal of Aerosol Science

4. The results of the optical emission spectroscopy seem very isolated from the rest of the publication. It is not clear how these experiments were performed. This is a bit confusing, is this supposed to be part of the experimental setup description, or already an experimental results section? If it is experimental results, then the procedure of how these measurements were done should be added in a bit more detail in the experimental section. Was this a completely separate measurement, or did you do that while aerosol generation and charging was happening as well? This would require also a description of the source itself, which was already mentioned above. Was the optical emission spectroscopy done only in pure helium environment, and how would that be relevant to the rest of the measurement?

Some more detailed comments:

Ad Section 2) Experimental Setup: No description, schematic or anything on the charger!

Page , Fig. 2: It seems like the mobility distributions were measured in an air Helium, Argon, or N2 mixture. But I don't see any discussion of the influence of this gas mixture on the mobility measurements. If the DMA has been operated in a closed loop this has to be considered. The mobility of THAB in a helium air mixture would not be the same, so how was the calibration done in this case? If this was considered and found to be negligible a discussion and reference has to be added. If it has not been considered, then this needs to be done.

Section 2.1. I think this should be numbered 3, not 2.1, since it does not seem to be part of the experimental setup

Page 5, line 94, 95: What is the copper antenna for?

Page 8, line 144 says that the different masses of charger ions created in the plasma torch and the other charger sources might result in the observed differences. Can this be quantified. Is the mean mass, and mobility higher or larger than in the other case. How does an increase or decrease of mass and mobility affect the final result. Why did you not apply the measured mass and mobility to the theory?

Page 9, line 163, why would it charge better in air than in helium? And how can the large difference of 50% be explained?

Page 10, line 185, polarities wrong, also, you should mention that it was the y-axis that was normalized

Page 11, Fig. 7, for negative mass graphs the chemical equations are mentioned but not the mass, while for the positive ions it's the other way around. Is there a reason for that? What are the rectangles in the positive Am-241?

---

## Author Comment (AC1) · 1 Jun 2020

We appreciate the thoughtful comments by referee 1. For discussion purposes we would like to respond to the general and detailed points raised.

*Manuscript by Tauber et al. reports characterization of a new bipolar non-thermal plasma type charger for charging sub-15 nm aerosol particles. The main results of the study are measured size dependent charging efficiencies that are strongly biased to negative polarity, and mass and mobility characterization of the charger ions. Experimental data on sub-10 nm charging efficiencies are in the general interest in aerosol measurement techniques, while the current*

[Figure]

*manuscript requires elaboration on several places prior to consideration for publication. General and specific comments are listed below*

*General:*

*- Experimental section: there is absolutely no description of the new plasma charger which is being characterized. This undermines the reproducibility of the most important results of the whole manuscript. The operation principle and general characteristics of the charger have to be described prior to publication. Otherwise a serious scientist in another lab cannot reproduce the potentially interesting results obtained with the charger. This applies also to the other used chargers.*

There is indeed not much information given about the charger itself and we acknowledge that this is unusual for a research paper. However, we have to point out that there is still a patent pending and, thus, we cannot reveal all the technical details of how the charger works. Broadly speaking the atmospheric pressure plasma charger consists of a gas flow that is shielded by another gas flow from the surrounding atmosphere. The plasma is ignited inside the inner flow while the aerosol is administered through the outer gas stream. The main source of the plasma is a high-frequency copper electrode that is situated on the central axis of those two gas streams.

We will add the following description to the experimental section:
"The atmospheric pressure plasma charger consists of a gas flow that is shielded by another gas flow from the surrounding atmosphere. The plasma is ignited inside the inner flow while the aerosol is administered through the outer gas stream. The main source of the plasma is a high-frequency copper electrode that is situated on the central axis of those two gas streams. According to Kallinger et al. (2012), the used radioactive 241Am charger has a cylindrical geometry with an axial flow direction. The radioactive source is mounted on the inner wall. The chamber has an inner diameter of about 30 mm and a length of 120 mm. Furthermore, the soft x-ray charger is composed of a

stainless-steel tube and a photo ionizer. The aerosol particles are directed along the tube towards the soft x-ray source and leave the charger via an outlet, that is oriented perpendicularly to the axis of the tube. The tube has an inner diameter of 30 mm and a length of 200 mm."

*- P8 L136 onward and P15 L223 onward, the authors discuss the effect of ion mass on the charging probability. They have even measured the ion masses and mobilities, while still in the theoretical prediction they use presumably "wrong" ion masses and mobilities. Why? Why not predicting the charging efficiencies with the measured ion mobility and mass? This is also speculated in the conclusions with statements that the charging efficiencies biased to negative polarity are explainable with the ion mobilities, while not a single ion mobility value (which should be actually weighed with the ion concentration) is reported even if they are measured.*

We would like to thank the reviewer for making us aware of that and add the following to results section:
"In Table 1R the measured and calculated mean ion charger mobilities, mobility equivalent diameters, masses and ion mobility ratios are listed. For comparison the values found by Reischl et al. (1996) are also listed. The results were used to calculate the charge distribution with Fuchs theory as shown in Figure 1. Negatively charged particles in the size range from 4-12 nm by 241Am or X-Ray neutralizers agree well with the parameters derived by Reischl et al. (1996) and an ion mobility ratio of 1. For positively charged particles the charging efficiency is below the measurement results for particles between 4-10 nm. By correcting the charge distribution with the parameters derived by Reischl et al. (1996) with an ion mobility ratio of 0.8 the negatively charged particles with a size below 4 nm fit perfectly to theory. The measurement results of this work reveal an increased charging efficiency for both polarities as shown in Figure 1. For mobility equivalent diameters between 4 and 12 nm and positive polarity the charging efficiency fits with theory for 241Am, X-Ray and the plasma torch with air as working

gas. In contrast to negatively charged particles where the results of the plasma torch with nitrogen or air as working gas above 7 nm is higher and below 7 nm is lower than expected by theory. Also, for 241Am, X-Ray and the plasma charger with helium the theory exceeds the measured charging efficiency. By correcting the theory with the acquired ion mobility ratio, a good agreement between theory and measurement can be found for negatively charged particles above 7 nm and for positively charged particles for the plasma torch with nitrogen as working gas. Although the effect for diameters > 7 nm can be explained, there is still a deviation for the smaller diameters from theory. The reported discrepancy can therefore not solely be attributed to the ion mobilities. There are other effects which should by investigated in further studies. Especially the charging effects below 5 nm which cause deviations from the charging model."

Table 1R. Comparison of ion cluster properties: polarity, mobility diameter Dp calculated from mean ion mobility Z, mean ion mass M and ion mobility ratio Z-/Z+.

|  | Polarity | Dp [nm] | Z [cm2/Vs] | M [Da] | $Z^-/Z^+$ | |
|---|---|---|---|---|---|---|
| Reischl et al. (1996) | + | 1.32 | 1.15 | 290 | 1.0 | 0.80 |
| Reischl et al. (1996) | - | 1.19 | 1.43 | 140 | 1.0 | 0.80 |
| measured | + | 1.07 | 1.76 | 356 | 1.0 | 0.66 |
| measured | - | 0.87 | 2.66 | 116 | 1.0 | 0.66 |

The following caption will be added to Figure 1:
"Measured charging efficiencies for the different aerosol chargers for negatively and positively charged Ag particles (dots) and negatively charged NaCl particles (squares) with mobility diameters less than 12 nm. The lines represent the charge distribution according to Fuchs theory, the parameters for the ion mobilities, ion masses and ion mobility ratio are listed in Table 1R. The collision probability of ions was calculated following Hoppel and Frick (1986)."

*- Generally there are several experimental details that are not reported in sufficient detail, or statements of which interpretation is ambiguous. Some of these*

***are picked below.***

***Specific:***

***- P1 L5, "increased charging efficiency", compared to what?***

The sentence will be corrected:
"A comparison of the different neutralization methods revealed an increased charging efficiency for negatively charged particles using the non-radioactive plasma charger with nitrogen as working gas compared to a radioactive americium neutralizer."

***- P1 L7, "charging mechanism", are you referring to the different neutralizers? Charging mechanism is a different thing***

Yes, we refer to the different neutralizers.
The sentence will be corrected:
"The mobility and mass spectrometric measurements show that the generated neutralizer ions are of the same mobilities and composition independent of the examined neutralizer."

***- P1 L8, in which sense the TSI X-ray is "standard" neutralizer? I would say commercial***

The sentence will be corrected:
"It was the first time that the Gilbert Mark I plasma charger was characterized in comparison to a commercial TSI X-Ray (TSI Inc, Model 3088) and a radioactive americium neutralizer."

***- P1 L9, "enhance" compared to what?***

The sentence will be corrected:
"We observed that the plasma charger with nitrogen as working gas can enhance the charging probability for sub-10 nm particles compared to a radioactive americium neutralizer."

*- P1 L10, "increased down towards", reformulate*

The sentence will be corrected:
"Consequently, the limit of detection of differential or scanning mobility particle sizers can be increased to nanometer sized particles with the Gilbert Mark I plasma charger."

*- P1 L20, "below 10 nm in diameter are typically difficult to neutralize", I would say the opposite*

The sentence will be corrected:
"Aerosol particles below 10 nm in diameter are typically difficult to charge and carry only one electrical charge at maximum (Wiedensohler, 1988)."

*- P1 L21, "Quantitative particle detection in this size range is extremely challenging due to high diffusional losses. Hence, a higher charging efficiency is of importance to improve the signal intensity in the sub-10 nm regime", the first sentence implies challenges in determining the diffusion losses, i.e. systematic uncertainty, while the second one deals with detection limit and poisson type random uncertainty. Improvements in the second does not help the first one.*

We thank the reviewer for making us aware of this unclear formulation and the sentences will be reformulated:
"Quantitative particle detection in this size range is extremely challenging due to high diffusional losses, which results in low number concentrations. Therefore, a higher charging efficiency is of importance to increase the detectable number concentration in the sub-10 nm regime."

*- P2 L33, it will be good to define what is non-thermal plasma, if it is the operation principle of the new charger*

We fully agree with the referee on this point and added the following sentences to the manuscript:
"The term non-thermal plasma is usually used to describe a discharge in which the

electrons are in thermal non-equilibrium with the ions. This means that the average temperature of the gas in such a discharge is far lower than the temperature of a thermal plasma (i.e. some hundred K compared to several thousand K in the latter case)."

*- P2 L39, "mobility spectrum of . . .", it is the mobility spectrum of the charger ions, not of the chargers.*

The sentence will be corrected:
"In the past, various studies have characterized the charging probabilities and mobility spectra of the charger ions produced by AC-corona, X-Ray or alpha-radiation based chargers (Wiedensohler et al., 1986; Steiner and Reischl, 2012; Kallinger et al., 2012; Kallinger and Szymanski, 2015)."

*- P2 L46, "steady-state", earlier you talked about equilibrium, which one it is?*

We measured the charging probability and assumed that the charge equilibrium inside the charger leads to a well-known size-dependent charge distribution. The chargers were measured during steady-state operational conditions. We will reformulate the sentence:
"Here we report on size-dependent charging probability measurements of a non-thermal plasma source (Gilbert Mark I plasma charger, Gruenwald Laboratories GmbH, Austria), an americium 241 (241Am) charger and of a TSI Advanced Aerosol Neutralizer 3088 by means of a tandem DMA (Differential Mobility Analyzer) setup as depicted in Figure 1."

*- Figure 1, what is the sheath the flow rate and size resolution of the nDMAs? What about the furnace output distribution? How much the size resolution of the first DMA affects the results? Based on the plot, the DMA2 was operated with aerosol flow of 8 lpm at some point, can it really handle such high flow? Is it so that only the Torch requires the working gas? It is not clear from the figure. If the torch mixes helium with the sample flow, it changes the gas composition.*

***Does it affect sizing of the DMA or the flow rates of the CPC? If yes, how was this accounted for?***

The ratios for sheath flow and aerosol flow for the nDMA are listed below:
sheath flow 19.5 lpm / aerosol flow 2.5 lpm = 7.8
sheath flow 33.0 lpm / aerosol flow 5.0 lpm = 6.6
sheath flow 41 lpm / aerosol flow 8.0 lpm = 5.1

The furnace output distribution is constant because the 3 lpm through the furnace were not changed. The additional flow was introduced after the aerosol size classification and the aerosol distribution was checked before a measurement run to reach the desired concentration for the analyzed particle size. Both nDMAs are of identical construction and the classifier (nDMA 1) was always operating at the same sheath flow ratio (19.5 lpm / 3 lpm = 6.5). Consequently, we assume negligible effect of the size resolution of the first nDMA.

The geometric standard deviation of the particle size for the used nDMAs was evaluated by Winkler et al. (2008) for a sheath flow of 25 lpm and an aerosol flow of 4.6 lpm and is below 1.05 for particles with a mobility diameter down to 2 nm. The resulting flow ratio (25 / 4.6 = 5.4) is close to our measurement with 8 lpm aerosol flow and the signal-to-noise ratio was comparable to the measurements with lower aerosol flow rates.

Only the Torch requires an additional working gas, we will add the following sentence to the figure caption: "The additional working gas flow is only needed for the plasma charger." The additional flow rate from the working gas is at max 1/9 of the aerosol flow. According to our measurements no sizing effects were observed. Since the generated charger ions are comparable, one can assume that the charged particles are comparable, and the sizing of the DMA is not affected. In addition, helium is an inert gas and therefore non-reactive and combine rapidly with the surrounding substances.

We will add the following paragraph in the experimental setup section:

"The geometric standard deviation of the particle size for the used nDMAs was evaluated by Winkler et al. (2008) for a sheath flow of 25 lpm and an aerosol flow of 4.6 lpm and is below 1.05 for particles with a mobility diameter down to 2 nm. The resulting flow ratio (sheath / aerosol = 5.4) is close to our measurement with 8 lpm aerosol flow and the signal-to-noise ratio was comparable to the measurements with lower aerosol flow rates. The different flow settings for the nDMAs are listed in Table 2R."

Table 2R. Flow rates for the aerosol and sheath flow for the used nDMAs with the calculated sheath flow ratio.

| nDMA | Aerosol flow [L/Min] | Sheath flow [L/Min] | Ratio |
|------|----------------------|---------------------|-------|
| 1 | 3.0 | 19.5 | 6.5 |
| 2 | 2.5 | 19.5 | 7.8 |
| 2 | 5.0 | 33.0 | 6.6 |
| 2 | 8.0 | 41.0 | 5.1 |

*- P2 L56, which kind of ion trap?*

The ion-trap used consists of two symmetrical half shells separated by an isolator. The charged ions and particles were removed by an ion-trap which was set to +/- 500 V. The capturing efficiency of the electrostatic precipitator was found to be $> 99\%$ in the size range between 1 and 10 nm (Brilke et al. 2020; Tauber et al., 2019b).

*- P5 L93-94, "It can be concluded from the OES spectra that there is no ionization of aerosol particles facilitated by the carrier gas, since only neutral helium emission lines have been recorded." It is not clear what is the experimental setup during this experiment. It is only mentioned that helium is fed to the charger, there should be no particles in helium in the first place?*

The optical emission spectrometer was located at the nozzle of the plasma charger and used to record spatially averaged optical data along the axis of the plasma source.

*- P5 L94, "atoms from the copper high frequency antenna", what is this antenna?*

We will add the following sentences on P5 L89 to explain the antenna and its usage: "Thereby the plasma jet is shielded by another gas flow from the surrounding atmosphere. The plasma is ignited inside the inner flow while the aerosol is administered through the outer gas stream. The main source of the plasma is a high-frequency copper antenna/electrode that is situated on the central axis of those two gas streams."

*- Section 2.1.1, what is the purpose of this section, or what we should conclude in terms of aerosol charging? Are the emission spectra comparable to the conditions of the charging efficiency experiments, or to the recorded mass spectra?*

The emission spectra yield additional information about the charging mechanism and the plasma itself. The former encompasses the discovery that the charging of the aerosol particles is achieved via electrons that origin from the central electrode of the charger. The latter point includes, for example, the detection of singly charged He particles, which have a lifetime that is so short that they recombine before reaching the detector in the ioniAPi-TOF. On the other hand, the optical emission spectroscopy (OES) measurements were done in order to compare this novel charging mechanism to established ones like the X-ray or Americium chargers. Furthermore, the OES measurements led to a better understanding of the plasma behavior and, thus, the way of charge transfer from the plasma to the molecules or aerosols. The electrons are detached from the high-frequency copper electrode as those atoms are easily ionized. After leaving the electrode the charges attach themselves onto the preexisting molecules or aerosol particles. As a result, based on the ioniAPi-TOF and mobility measurements it can be shown that the different charging mechanism lead to comparable measurement results.

*- P6 L103, "reduced temperature settings", what were the settings? - P6 L107 "In Tauber et al. (2019b) the particle counting efficiency of the CPCs used here was determined, and the results obtained were corrected for the CPC detection efficiency", it is not clear what are used settings, since in the reference there are three different settings for the CPC.*

We will change the following sentence and include the used temperature settings:
"Two butanol-based CPCs (TSI 3776 UCPC) with reduced temperature settings (Condenser 1.1°C, Saturator 30.1°C, Optics 31.1°C) compared to factory settings to increase the particle counting efficiency were used (Barmpounis et al., 2018; Tauber et al., 2019a)."

*- P6 L100-110, the description of the experiment should be in section 2 where the setup is described. How what about losses in the dilution, how they were corrected? The second DMA scans the voltage, what is the CPC2 signal that is compared to the CPC1?*

We will move P6 L100-110 to the experimental description section and exchange Figure 1 with the Figure 2 because of a misleading plot. The number concentration measurements were done after the dilution and not before as shown in the previous plot. The number concentration measurements of CPC 2 were inverted to calculate the number size distribution from the mobility distribution according to Petters et al. (2018), which was then compared to the average particle concentration measured by CPC 1.

*- P6 L120, what the particle concentration has to do with charging efficiency? At extremely high concentrations concentration may affect the charging efficiency but not at low concentrations. Which CPC records the concentration that is mentioned? It is stated that data between 2-4 nm is in agreement with theory. Then the next sentence states this is not true for NaCl particles. Which one is true? Please elaborate the whole paragraph*

We will change the following paragraph accordingly:
"As the particle concentrations for very small diameters between 2-4nm are below 10000 /ccm, the charging efficiencies of the different devices vary strongly at these particle sizes. The nDMA in this size range has a low transmission efficiency ranging from $20 - 55\%$ and therefore the signal at the CPC 2 is very low. In addition, small temperature-fluctuations in the tube furnace lead to bigger uncertainties for the low total number concentration of the selected particle size. Especially for NaCl particles a big variation in the charging efficiencies of the different charging devices was observed. Those variations almost certainly are caused by the low number concentrations of NaCl particles at these sizes compared to Ag particles. However, except for NaCl particles at particle diameters between 2-4nm the associated data points for the 241Am neutralizer agree well with the approximations by Tigges et al. (2015) and Wiedensohler (1988)."

*- P7 L129, agrees with what?*

We will modify this sentence accordingly:
"Evidently, the data using helium agree well with theoretical approximations from Tigges et al. (2015) and Wiedensohler (1988) in the size regime between 4 and 10 nm."

*- P7 L131, what is dependent on what?*

We will modify this sentence accordingly:
"The data suggest no dependency of the charging efficiency on the charger ion polarity, when helium is used as working gas in the plasma torch."

*- P8 L137 onward, also charger ion mobility should affect the charging probability.*

Thank you for making us aware of this inconsistence, we change this sentence as follows:
"Wiedensohler and Fissan (1991) have shown that the predicted charging probabilities of NaCl and Ag particles strongly depend on the used carrier gas and the ion mass and mobility."

*- P8 L143 "As Figure 3 shows, the plasma torch forms copper ions and free electrons which charge aerosol particles in the carrier gas." Why these are not observed in the mass spectrum?*

The ions and free electrons are not observed in the mass spectrometer because their life time is far too short to reach the detector. For example, the transition probability of the ionization state represented by the copper emission line at 775.40 nm is $9 \times 10^6/s$ (Kramida et al., 2017). Thus, the mean lifetime of this ion is $1/(9 \times 10^6) \approx 0.1s$. Even if we assume a close to sonic flow velocity of the gas as an upper limit, the maximum range of the copper ion would be about $300m/s \times 10^{-6}s = 0.03mm$ before it recombines.

*- P8 L147, "the ions can form in a pure nitrogen environment almost like in the mentioned case of Wiedensohler and Fissan (1991)", what do you mean?*

We will modify this sentence accordingly:
"As the working gas flow is exposed to a high frequency electrical field before it mixes with the aerosol flow, the ions can form in a pure nitrogen environment. Hence the charger ions form in a nitrogen atmosphere like in the case of Wiedensohler and Fissan (1991) where silver and sodium chloride particles were charged with a Kr 85 source in a pure nitrogen atmosphere. "

*- Figure 5. It is not described how the normalization is done for the right hand side plot. It is hard to believe that the differences in the distributions are only due to charging. At 8-10 nm the charging efficiency of the X-ray is about 10 times larger than of the torch which is opposite to what is reported in Fig4. On the other hand, at 3 nm the torch shows charging efficiency of a factor of 5 higher than the X-ray, which is also contradictory to fig4.*

At this point we want to mention that Figure 5 is rather a qualitative representation than a quantitative one, because of the rotated periodical measurements which were conducted in multiple cycles. The discussed shifts in the number concentration especially for the small sizes, for example 3 nm particles might also be a result of the temperature-fluctuations inside the tube furnace. Furthermore, the deviation at 8-10 nm for the X-Ray charger might be a result of the ammonium sulfate contaminations measured

with the mass spec and by the ion mobility measurements where an increase of about 0.25 nm for the negative ions was found. The comparison of the mobility equivalent diameters from the classifier nDMA to the scanning nDMA did not show a significant difference for the 241Am neutralizer and the plasma torch.

We will change the following paragraph accordingly:
"Figure 5 depicts the recorded negatively charged aerosol size distribution averaged over numerous measurements, as well as the charging efficiency of the individual chargers normalized to the 241Am charger recordings. The normalization was done by comparing the recorded total number concentration of the individual neutralizers with the recorded total number concentration of the 241Am charger. The inversion of the size distribution data was performed according to Petters (2018). These diagrams permit a qualitative descriptive comparison of the different charging devices. The left plot reveals a shift of about 1 nm in the maximum of the recorded size distribution between the X-Ray and the plasma charger, which might be a result of small shifts in the size distribution originated from the tube furnace because of small temperature-fluctuations. Furthermore, contaminations of the X-Ray charger from previous experiments with ammonium sulfate might also lead to an increased particle diameter for the observed neutralizer (Steiner and Reischl, 2012)."

Furthermore, will change the Figure 5 caption:
"The plot on the left side represents the same recorded negative Ag aerosol size distribution with a UCPC, depending on the mobility diameter for the plasma torch (N2), X-Ray and americium neutralizers. On the right-hand side all conducted measurements with the same aerosol distribution are normalized to the 241Am results for a direct comparison."

*- Figure 6, the intro discusses with one sentence the nT product. Is that product sufficient in the chargers to reach the steady state charge distribution?*

According to the manufacturer the applied X-Ray neutralizer can be used for aerosol

flow rates up to 5 L/Min and the 241Am charger used in this study was characterized by Kallinger et al. (2015) for flowrates up to 5 L/Min. Hence, we used the 241Am neutralizer as a reference which reaches steady state charge distributions and compared it the other measurements. The higher charging efficiency might be a result of the high ion concentrations of about $10^{13}/cc$ which can be achieved in a cold plasma (Kurake et al., 2016). That is approximately 6 orders of magnitude above the ion concentration inside a conventional Am241 aerosol charger (Steiner et al., 2014).

*- P10 L185, ". The positive mass spectra were normalized to the nitrate ion (NO−3) peak at an integer mass of 62 Th and the negative mass spectra to the (H2O)2 H3O+ water cluster at an integer mass of 55 Th", the polarities do not match*

We would like to thank the reviewer for making us aware, we will change it in the manuscript accordingly:
"The negative mass spectra were normalized to the nitrate ion (NO−3) peak at an integer mass of 62 Th and the positive mass spectra to the (H2O)2 H3O+ water cluster at an integer mass of 55 Th."

*- Section 2.1.3, previously there was discussion on the ionization of copper, why there is no copper signal in the charger ions?*

Due to the low mean lifetime of this copper ions (about $1/(9 \times 10^6) \approx 0.1s$) it cannot be detected with the mass spec or with the UDMA setup.

*- P12 L211, "(NH4)14SO−4", is it really a cluster of 14 NH4 ammonium molecules?*

We will change the following sentence:
"According to our mass spec analysis, this is due to ammonium sulfate contaminations from previous experiments."

*- P15 L214, I would not say the are the same. Especially in positive there seem*

[Figure]

***to be differences***

We will reformulate this sentence:
"The presented measurements conducted with a non-thermal plasma source have shown that helium, nitrogen and air as working gases lead to the same ion species. According to the mobility and mass spec measurements the comparison of the plasma charger with the americium neutralizer indicates the same negative ion species. Whereas for the positive ions the measurements reveal a slight deviation."

***- P15 L218, it was shown nowhere that the torch produces ozone.***

Preliminary measurements were already conducted with an ozone monitor which will be included in the manuscript to support the statement in the conclusion section.

***- P15 L219, increased compared to what?***

We will change the following sentence accordingly:
"By switching the working gas to nitrogen an increased charging efficiency could be recorded for negatively charged particles compared to 241Am neutralizer."

References:
1) Barmpounis, K., A. Ranjithkumar, A. Schmidt-Ott, M. Attoui, and G. Biskos. Enhancing the detection efficiency of condensation particle counters for Sub-2nm particles. J. Aerosol Sci. 117 :44–53. doi: 10.1016/j.jaerosci.2017.12.005, 2018.
2) Sophia Brilke, Julian Resch, Markus Leiminger, Gerhard Steiner, Christian Tauber, Peter J. Wlasits  Paul M. Winkler. Precision characterization of three ultrafine condensation particle counters using singly charged salt clusters in the 1–4 nm size range generated by a bipolar electrospray source, Aerosol Science and Technology, 54:4, 396-409, DOI: 10.1080/02786826.2019.1708260, 2020.
3) William A. Hoppel  Glendon M. Frick. Ion-Aerosol Attachment Coefficients and the Steady-State Charge Distribution on Aerosols in a Bipolar Ion Environment, Aerosol Science and Technology, 5:1, 1-21, DOI: 10.1080/02786828608959073, 1986.

4) Kallinger, P. and Szymanski, W.: Experimental determination of the steady-state charging probabilities and particle size conservation in non-radioactive and radioactive bipolar aerosol chargers in the size range of 5–40 nm, Journal of Nanoparticle Research, 17, https://doi.org/10.1007/s11051-015-2981-x, 2015.

5) Kallinger, P., Steiner, G., and Szymanski, W.: Characterization of four different bipolar charging devices for nanoparticle charge conditioning, Journal of Nanoparticle Research, 14, https://doi.org/10.1007/s11051-012-0944-z, 2012.

6) Kramida, Alexander; Nave, Gillian; Reader, Joseph. The Cu II Spectrum. Atoms 5, no. 1: 9. doi:10.3390/atoms5010009, 2017.

7) Kurake N, Tanaka H, Ishikawa K.: Cell survival of glioblastoma grown in medium containing hydrogen peroxide and/or nitrite, or in plasma-activated medium. Arch Biochem Biophys. 605:102‐108. doi:10.1016/j.abb.2016.01.011, 2016.

8) Petters, M. D.: A language to simplify computation of differential mobility analyzer response functions, Aerosol Science and Technology,52, 1437–1451, https://doi.org/10.1080/02786826.2018.1530724, 2018.

9) Reischl, G., Mäkelä, J., Karch, R., and Necid, J.: Bipolar charging of ultrafine particles in the size range below 10 nm, Journal of Aerosol Science, 27, 931 – 949, https://doi.org/https://doi.org/10.1016/0021-8502(96)00026-2, fuchs Memorial Issue, 1996.

10) Steiner, G. and Reischl, G. P.: The effect of carrier gas contaminants on the charging probability of aerosols under bipolar charging conditions, Journal of Aerosol Science, 54, 21 – 31, https://doi.org/https://doi.org/10.1016/j.jaerosci.2012.07.008, 2012.

11) Gerhard Steiner, Tuija Jokinen, Heikki Junninen, Mikko Sipilä, Tuukka Petäjä, Douglas Worsnop, Georg P. Reischl Markku Kulmala: High-Resolution Mobility and Mass Spectrometry of Negative Ions Produced in a 241Am Aerosol Charger, Aerosol Science and Technology, 48:3, 261-270, DOI: 10.1080/02786826.2013.870327, 2014.

12) Tauber, C., Brilke, S., Wlasits, P., Bauer, P., Köberl, G., Steiner, G., and Winkler, P.: Humidity effects on the detection of soluble and insoluble nanoparticles in butanol

operated condensation particle counters, Atmospheric Measurement Techniques, 12, 3659–3671,https://doi.org/10.5194/amt-12-3659-2019, 2019a.

13) Tauber, C., Steiner, G., and Winkler, P. M.: Counting efficiency determination from quantitative intercomparison be-tween expansion and laminar flow type condensation particle counter, Aerosol Science and Technology, 53, 344–354,https://doi.org/10.1080/02786826.2019.1568382, 2019b.

14) Tigges, L., Wiedensohler, A., Weinhold, K., Gandhi, J., and Schmid, H.-J.: Bipolar charge distribution of a soft X-ray diffusion charger, Jour-nal of Aerosol Science, 90, 77 – 86, https://doi.org/https://doi.org/10.1016/j.jaerosci.2015.07.002, 2015.

15) Wiedensohler, A.: An approximation of the bipolar charge distribution for particles in the submicron size range, Journal of Aerosol Science, 19, 387 – 389, https://doi.org/https://doi.org/10.1016/0021-8502(88)90278-9, 1988.

16) Wiedensohler, A. and Fissan, H. J.: Bipolar Charge Distributions of Aerosol Particles in High-Purity Argon and Nitrogen, Aerosol Science and Technology, 14, 358–364, https://doi.org/10.1080/02786829108959498, 1991.

17) Wiedensohler, A., Lütkemeier, E., Feldpausch, M., and Helsper, C.: Investigation of the bipolar charge distribution at various gas conditions, Journal of Aerosol Science, 17, 413 – 416, https://doi.org/https://doi.org/10.1016/0021-8502(86)90118-7, 1986.

18) Winkler, P. M., G. Steiner, A. Vrtala, H. Vehkamäki, M. Noppel, K. E. J. Lehtinen, G. P. Reischl, P. E. Wagner, and M. Kulmala. Heterogeneous nucleation experiments bridging the scale from molecular ion clusters to nanoparticles. Science 319 (5868):1374–1377. doi: 10.1126/science.1149034, 2008.

Please also note the supplement to this comment:
https://www.atmos-meas-tech-discuss.net/amt-2020-54/amt-2020-54-AC1-supplement.pdf
* * *
[Figure]

Fig. 1. The charge distribution according to Fuchs theory, the parameters for the ion mobilities, ion masses and ion mobility ratio are listed in Table 1R.

[Figure]

**Fig. 2.** Schematic of the experimental setup for the charging probability and particle size distribution inversion measurements. The additional working gas flow is only needed for the plasma charger.

---

## Author Comment (AC2) · 22 Jun 2020

We appreciate the thoughtful comments by referee 2. For discussion purposes we would like to respond to the general and detailed points raised.

*This work describes the use of a plasma torch as an aerosol neutralizer. The work measures and compares the charging probability of the new source with commercial available other aerosol chargers. The charging probabilities were measured for positive and negative particles (Ag, and NaCl) and at different aerosol flow rates. In addition, the plasma torch has been evaluated at operation with different working fluids (He, Ar, N2). The mobility distributions and mass*

*graphs of the charger ions were also measured in order to get an information on the properties of the charger ions. The work includes a very thorough investigation of the charger source and the charger ions. However, there are a few general concerns about the work, that should be addressed before publication:*

*1. The plasma torch itself, is not described at all in the current manuscript, a paragraph on the working principle should be added to the manuscript.*

There is indeed not much information given about the charger itself and we acknowledge that this is unusual for a research paper. However, we have to point out that there is still a patent pending and, thus, we cannot reveal all the technical details of how the charger works. Broadly speaking the atmospheric pressure plasma charger consists of a gas flow that is shielded by another gas flow from the surrounding atmosphere. The plasma is ignited inside the inner flow while the aerosol is administered through the outer gas stream. The main source of the plasma is a high-frequency copper electrode that is situated on the central axis of those two gas streams.

We will add the following description to the experimental section:
"The atmospheric pressure plasma charger consists of a gas flow that is shielded by another gas flow from the surrounding atmosphere. The plasma is ignited inside the inner flow while the aerosol is administered through the outer gas stream. The main source of the plasma is a high-frequency copper electrode that is situated on the central axis of those two gas streams."

*2. In the measurement with different gases, one would expect that using a DMA and CPC in a helium-air mixture would result into changes in the instrument performance. The voltage mobility relationship in the DMA is gas dependent, and in the CPC the flow calibration would change when adding helium to the system, also the supersaturation profile would change and the detection limit would shift to smaller particles (e.g. Thomas et al. 2018, Journal of Heat and Mass Transfer). These are points that should be addressed in this manuscript.*

We will add the following statement to the manuscript:
"The additional flow rate from the working gas was at max 1/9 of the aerosol flow. According to Thomas et al. (2018) a cutoff drift to lower sizes for helium mole fractions below 0.67 was found for butanol-based CPCs. However, the used CPC in this study was operated with reduced temperature settings and thereby a lower detection efficiency was established (Tauber et al. 2019a). As a result, the recorded cutoff drift would therefore only influence the charging efficiency measurements conducted at < 3 nm. The resulting error is already covered for this particle sizes by the measurement uncertainties of nDMA and CPC."

*3. The charger ion mass and mobilities were measured, however, no qualitative thoughts were included in how and why that would result into the observed changes of charging probability. It has been stated correctly that the charger ion composition plays an important role to the final charge distribution. But what is missing is to apply the information found in this work to existing theories and see if the trend agrees with the observations. Simulation results of charge distributions considering different ion mass and mobility of charger ions have been performed in the past, see for example Maisser et al, 2015, Journal of Aerosol Science*

We compared our results to approximations given by Wiedensohler (1988) and Tigges et al. (2015) and performed calculations for different ion masses and mobilities. The results of the calculations are posted in the review comment 1 and will be added to the manuscript.

*4. The results of the optical emission spectroscopy seem very isolated from the rest of the publication. It is not clear how these experiments were performed. This is a bit confusing, is this supposed to be part of the experimental setup description, or already an experimental results section? If it is experimental results, then the procedure of how these measurements were done should be added in a bit more detail in the experimental section. Was this a completely separate*

***measurement, or did you do that while aerosol generation and charging was happening as well? This would require also a description of the source itself, which was already mentioned above. Was the optical emission spectroscopy done only in pure helium environment, and how would that be relevant to the rest of the measurement?***

The optical emission spectroscopy was conducted as a separate experiment with the flow rates stated in the supplemental material but without aerosol generation. The emission spectra yield additional information about the charging mechanism and the plasma itself. The former encompasses the discovery that the charging of the aerosol particles is achieved via electrons that originate from the central electrode of the charger. The latter point includes, for example, the detection of singly charged He particles, which have a lifetime that is so short that they recombine before reaching the detector in the ioniAPi-TOF. Furthermore, the OES measurements led to a better understanding of the plasma behavior and, thus, the way of charge transfer from the plasma to the molecules or aerosols. The electrons are detached from the high-frequency copper electrode as those atoms are easily ionized. After leaving the electrode the charges attach themselves onto preexisting molecules or aerosol particles. As a result, based on the ioniAPi-TOF and mobility measurements it was shown that the different charging mechanism lead to comparable measurement results.

We will add the following to the experimental section:
"The optical emission spectrometer was located at the nozzle of the plasma charger and used to record spatially averaged optical data along the axis of the plasma source."

***Some more detailed comments: Ad Section 2) Experimental Setup: No description, schematic or anything on the charger!***

We will add the following description to the experimental section:
"The atmospheric pressure plasma charger consists of a gas flow that is shielded by another gas flow from the surrounding atmosphere. The plasma is ignited inside the

inner flow while the aerosol is administered through the outer gas stream. The main source of the plasma is a high-frequency copper electrode that is situated on the central axis of those two gas streams. According to Kallinger et al. (2012), the used radioactive 241Am charger has a cylindrical geometry with an axial flow direction. The radioactive source is mounted on the inner wall. The chamber has an inner diameter of about 30 mm and a length of 120 mm. Furthermore, the soft x-ray charger is composed of a stainless-steel tube and a photo ionizer. The aerosol particles are directed along the tube towards the soft x-ray source and leave the charger via an outlet, that is oriented perpendicularly to the axis of the tube. The tube has an inner diameter of 30 mm and a length of 200 mm."

*Page, Fig. 2: It seems like the mobility distributions were measured in an air Helium, Argon, or N2 mixture. But I don't see any discussion of the influence of this gas mixture on the mobility measurements. If the DMA has been operated in a closed loop this has to be considered. The mobility of THAB in a helium air mixture would not be the same, so how was the calibration done in this case? If this was considered and found to be negligible a discussion and reference has to be added. If it has not been considered, then this needs to be done.*

The mobility distributions were measured with air as carrier gas and only for the plasma charger an additional working gas (Air, N2, He) was added. This working gas flow was between 40 and 280 cc/min. The mobility spectrum / calibration measurement with THAB was always recorded with air as carrier gas. After the calibration the charger was mounted to the setup and the experiments with different working gases was conducted. So, there was no helium air mixtures during the calibration runs in the sheath air of the UDMA.

*Section 2.1. I think this should be numbered 3, not 2.1, since it does not seem to be part of the experimental setup*

Thank you for making us aware of the wrong numbering. We will separate the results

and discussion section from the experimental section.

**Page 5, line 94, 95: What is the copper antenna for?**

We will add the following sentences on P5 L89 to explain the antenna and its usage:
"Thereby the plasma jet is shielded by another gas flow from the surrounding atmosphere. The plasma is ignited inside the inner flow while the aerosol is administered through the outer gas stream. The main source of the plasma is a high-frequency copper antenna/electrode that is situated on the central axis of those two gas streams."

**Page 8, line 144 says that the different masses of charger ions created in the plasma torch and the other charger sources might result in the observed differences. Can this be quantified. Is the mean mass, and mobility higher or larger than in the other case. How does an increase or decrease of mass and mobility affect the final result. Why did you not apply the measured mass and mobility to the theory?**

We would like to thank the reviewer for his thoughtful comments and make him aware of the performed calculations for different ion masses and mobilities which are posted in the review comment 1 and which will be added to the manuscript.

**Page 9, line 163, why would it charge better in air than in helium? And how can the large difference of 50% be explained?**

We will add the following paragraph to the manuscript:
"According to Maisser et al. (2015), nitric acid has an anomalously high gas phase acidity for its mass and can persist in the gas phase in higher concentrations than other low mass species. By using helium as working gas the concentration of nitrate ions in the gas phase is lower than in air or $N_2$ and therefore charge transport decreases. This is contrary to using $N_2$ as working gas where an increased charging efficiency up to 50% was measured."

**Page 10, line 185, polarities wrong, also, you should mention that it was the y-**

*axis that was normalized*

We will change it in the manuscript accordingly:
"The negative mass spectra were normalized to the nitrate ion (NO−3) peak at an integer mass of 62 Th and the positive mass spectra to the (H2O)2 H3O+ water cluster at an integer mass of 55 Th."

***Page 11, Fig. 7, for negative mass graphs the chemical equations are mentioned but not the mass, while for the positive ions it's the other way around. Is there a reason for that? What are the rectangles in the positive Am-241?***

The dashed square box marks unidentified masses in the positive 241Am mass spectrum and the solid square box shows the silicone compounds that are listed in Table 2 in the Manuscript. For space reason we mentioned only the chemical equations for the negative masses but in Table 3 all chemical equations and masses are listed.

***References:***

1. Kallinger, P., Steiner, G., and Szymanski, W.: Characterization of four different bipolar charging devices for nanoparticle charge conditioning, Journal of Nanoparticle Research, 14, https://doi.org/10.1007/s11051-012-0944-z, 2012.
2. Maisser, A., Thomas, J. M., Larriba-Andaluz, C., He, S., Hogan, C. J., The mass-mobility distributions of ions produced by a Po-210 source in air. Journal of Aerosol Science, 90, 36-50. https://doi.org/10.1016/j.jaerosci.2015.08.004, 2015.
3. Tauber, C., Steiner, G., and Winkler, P. M.: Counting efficiency determination from quantitative intercomparison be-tween expansion and laminar flow type condensation particle counter, Aerosol Science and Technology, 53, 344–354,https://doi.org/10.1080/02786826.2019.1568382, 2019a.
4. Jikku M. Thomas and Xiaoshuang Chen and Anne Maißer and Christopher J. Hogan, Differential heat and mass transfer rate influences on the activation efficiency of laminar flow condensation particle counters. International Journal of Heat and Mass Transfer, 127, 740-750. https://doi.org/10.1016/j.ijheatmasstransfer.2018.07.002,

2018.

5. Tigges, L., Wiedensohler, A., Weinhold, K., Gandhi, J., and Schmid, H.-J.: Bipolar charge distribution of a soft X-ray diffusion charger, Jour-nal of Aerosol Science, 90, 77-86, https://doi.org/https://doi.org/10.1016/j.jaerosci.2015.07.002, 2015.

6. Wiedensohler, A.: An approximation of the bipolar charge distribution for particles in the submicron size range, Journal of Aerosol Science, 19, 387 – 389, https://doi.org/https://doi.org/10.1016/0021-8502(88)90278-9, 1988.

Please also note the supplement to this comment:
https://www.atmos-meas-tech-discuss.net/amt-2020-54/amt-2020-54-AC2-supplement.pdf